# Reuse your FLOPs: Scaling RL on Hard Problems by Conditioning on Very Off-Policy Prefixes

Amrith Setlur [1] [*]  Zijian Wang [2]  Andrew Cohen [2]  Paria Rashidinejad [3] [*] [†]  Sang Michael Xie [2] [*] [†]

## Abstract

Typical reinforcement learning (RL) methods for LLM reasoning waste compute on hard problems, where correct *on-policy* traces are rare and policy gradients vanish. To bootstrap more efficient RL, we consider reusing old sampling FLOPs (from prior inference or RL training) in the form of *off-policy* traces. We introduce **PrefixRL**, where we *condition* on the prefix of successful off-policy traces and run on-policy RL to complete them, side-stepping instabilities from using off-policy data as supervision targets. PrefixRL boosts the learning signal on hard problems by modulating the difficulty of the problem through the off-policy prefix length. We prove that the PrefixRL objective is not only consistent with the standard RL objective but also more sample efficient. Empirically, we discover **back-generalization**: training *only* on prefixed problems generalizes to *out-of-distribution* unprefixed performance, with learned strategies often differing from those in the prefix. In our experiments, we source the off-policy traces by rejection sampling with the base model, creating a self-improvement loop. On hard reasoning problems, PrefixRL reaches the same training reward $2\times$ faster than the strongest baseline (SFT on off-policy data then RL), even after accounting for the compute spent on the initial rejection sampling, and increases the final reward by $3\times$.

## 1. Introduction

Reinforcement learning (RL) is the de facto method to boost large language model (LLMs) reasoning, especially for math and coding (An et al., 2025; Liu et al., 2025b; Guo et al., 2025). Most successful RL recipes (Ahmadian et al., 2024; Yu et al., 2025) are *on-policy*: sample multi-

---
[*]Work done at Meta. [†]Equal advising. [1]Carnegie Mellon University [2]Meta [3]University of Southern California. Correspondence to: Amrith Setlur <asetlur@andrew.cmu.edu>.

*Proceedings of the 43ʳᵈ International Conference on Machine Learning*, Seoul, South Korea. PMLR 306, 2026. Copyright 2026 by the author(s).

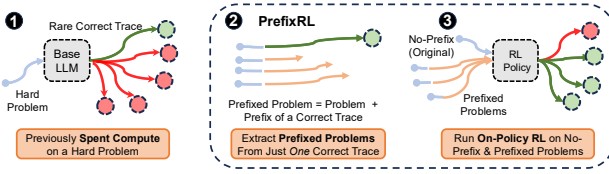

*Figure 1.* **PrefixRL: On-Policy RL Conditioned on Off-Policy Prefixes.** We leverage previously spent compute (❶) on hard problems in the form of correct off-policy traces obtained via rejection sampling with the base LLM we start RL from (trace could also come from other model families or previous RL runs). We append prefixes of a single correct off-policy trace per problem to the original problem, creating prefixed problems (❷). Then, we run on-policy RL on prefixed and no-prefix (original) problems (❸). PrefixRL places the policy in high-reward states which boosts the learning signal. Performance transfers from the prefixed to no-prefix problems via a phenomenon we call back-generalization.

ple reasoning traces (rollouts) from the current model and derive updates from correct (and incorrect) traces. This paradigm breaks down on *hard problems* with low pass@$k$ (e.g., pass@2000≈0), where the model rarely samples a correct trace. In this regime, RL spends tons of FLOPs without receiving any learning signal, and RL rewards plateau.

In practice, we are rarely solving these problems for the first time since earlier RL runs or inference on previous models may have spent compute on the same (or similar) hard problems. The question is how to reuse this growing dataset of *off-policy* traces that may contain some correct traces even for hard problems to accelerate on-policy RL.

A straightforward approach is to treat the off-policy traces as supervision: perform supervised fine-tuning (a.k.a., mid-training) on the correct off-policy traces followed by standard on-policy RL (Wang et al., 2025d). However, SFT on a small set of correct traces can lead to memorization (Chu et al., 2025) and entropy collapse, which hurts exploration during subsequent RL (Zhang et al., 2025a). Alternatively, we can use off-policy traces directly in RL via importance weighting, but this is often unstable due to high-variance gradient estimates (Liu et al., 2025a; Yan et al., 2025). Both options use off-policy traces as target supervision, and since these off-policy traces are very low probability under the RL policy, this leads to suboptimal RL optimization.

To avoid these pitfalls, we propose **PrefixRL**: run on-policy RL *conditioned* on prefixes of correct off-policy traces instead of supervising on them (Figure 1). *First*, we extract few fixed off-policy prefixes and append them to the original problem to create *prefixed problems*. *Second*, we run on-policy RL on both *no-prefix* (original) problems and prefixed problems, where gradients are masked on the off-policy prefix. The off-policy prefixes place the RL policy in states that are more likely to succeed on hard problems, reducing gradient variance and enhancing the learning signal.

However, it is not immediately clear what the effect on the bias is. In Section D.1, we prove that when the prefixes are correct and realizable in the model class, (i) maximizers of the PrefixRL objective also maximize performance on the standard RL objective; and (ii) since the prefixes lessen the exploration burden, PrefixRL reduces suboptimality gap with fewer samples compared to standard RL (by a factor of context length). Overall, PrefixRL changes the on-policy RL objective by using off-policy prefixes solely to guide exploration and unblock training on hard problems.

**Back-generalization.** Beyond the theory, we empirically find an additional phenomenon behind the gains in PrefixRL we call *back-generalization*, where on-policy RL on *only* prefixed problems substantially boosts test performance on the original no-prefix problems, which were never trained on. Beyond the generalization in the face of train/test mismatch, back-generalization is distinctive for two reasons. First, back-generalization is a type of *generalization via shared parameters* because it alters the next-token distribution on prefixes it was never trained on (impossible in tabular RL). Second, we find that *back-generalization can be even more powerful than standard generalization* in RL (transfer across related problems). We show this in an in-context learning setup, where we run RL on problems prefixed with another problem and reasoning trace in context. We find that training on a problem P1 conditioned on a related problem P2 in context improves generalization from P1 to P2 substantially more than directly running RL on P1 (see Section 4.3).

**PrefixRL can discover and learn strategies beyond what is provided in the prefix.** Interestingly, the model does not simply back-generalize by imitating the off-policy prefix. Through controlled experiments, we find that PrefixRL is more compute efficient than standard RL at amplifying successful strategies and rejecting suboptimal ones, *even* when the suboptimal strategy is explicitly present in the off-policy prefix. We hypothesize that PrefixRL allows the model to more quickly identify the flaws in the suboptimal strategy and use this insight to find a better strategy (see Section 4.2).

**PrefixRL improves both compute efficiency and final performance.** In our experiments, we instantiate PrefixRL in a *self-improvement setting* by collecting a dataset of off-policy traces rejection sampled from the base model. On

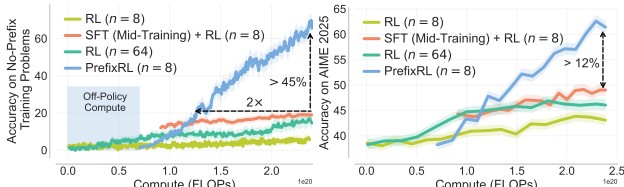

*Figure 2.* **PrefixRL affords a self-improvement pipeline that recycles FLOPs on hard problems.** We instantiate PrefixRL for self-improvement by collecting a dataset of off-policy traces through large-scale rejection sampling on the base LLM (distilled Llama3.1-8B). In FLOPs-matched training, PrefixRL outperforms the strongest baseline (SFT on off-policy data + RL): 2× higher compute-efficiency (including rejection sampling cost) and >45% higher final training accuracy on no-prefix training problems *(left)*, with gains transferring to standardized test benchmarks *(right)*.

hard problems in training, PrefixRL improves compute efficiency over the strongest mid-training baseline (SFT on the off-policy data then on-policy RL) by 2×, even when accounting for the compute spent on rejection sampling, and training accuracy by > 45% on the original no-prefix problems (Figure 2 *(left)*). These gains transfer to held-out benchmarks: *e.g.*, on AIME '25, PrefixRL improves pass@1 by 12% over the strongest mid-training baseline in a compute-matched comparison. Finally, we find that PrefixRL is still effective when off-policy prefixes are sourced from Qwen3-4B-instruct while the RL policy is a distilled Llama-3.1-8B-instruct. This setting provides similar compute and accuracy gains, demonstrating the flexibility of PrefixRL to the off-policy data source and model size.

## 2. Preliminaries

**Setup.** We use $x$ to denote a problem and $y = (y_1, \ldots, y_H)$ for a response of $H$ tokens sampled auto-regressively from an LLM or policy $\pi$ in class $\Pi$, where $y_{:h}$ refers to the prefix consisting of first $h$ tokens in $y$. We use $\pi^0$ to denote the base (pre-trained) LLM that we want to post-train on a dataset of hard problems $\mathcal{D} =: \{x_i\}_{i=1}^{|\mathcal{D}|}$ with verifiable rewards. We have an outcome reward $r(x_i, y)$ which is 1 when $y$ is correct and 0 when incorrect. We say $x$ is a *hard problem* for $\pi^0$ if pass@512 under $\pi^0$ is $\approx 0$ for $x$. We use $J(\pi) := \mathbb{E}_{x \sim \rho} \mathbb{E}_{y \sim \pi(\cdot|x)} r(x, y)$ to denote the performance of $\pi$ on $\rho$ which is the empirical distribution over $\mathcal{D}$. For full set of notations, see Appendix A. Our goal is to train $\pi^0$ on $\mathcal{D}$, to maximize $J(\pi)$ with access to previously spent compute on $\pi^0$ (or on models fine-tuned from it), available in the form of correct off-policy traces $\mathcal{D}_{\text{off}}$.

**Source of off-policy traces.** To situate ourselves in a *self-improvement* setup, we mainly source the off-policy traces via rejection sampling on the base model $\pi^0$. Concretely, for each $x \in \mathcal{D}$, we collect a single correct trace by running rejection sampling on $\pi^0$ until we see a correct trace. Therefore, if the pass@1 under $\pi^0$ is $p_x$ on $x$, then in expectation we will sample $1/p_x$ traces to get a correct one. Doing this for every $x \in \mathcal{D}$ gives us $\mathcal{D}_{\text{off}}$, where $|\mathcal{D}_{\text{off}}| = |\mathcal{D}|$. In

theory, we assume that the empirical distribution defined by $\mathcal{D}_{\text{off}}$ can be *realized* (perfectly fitted) by some $\mu \in \Pi$. In practice, $\mathcal{D}_{\text{off}}$ can also be curated with sophisticated inference algorithms that may not be representable by using greater compute depth than the architecture (by scaffolding sequential and parallel compute) or having oracle access. We show the flexibility of the off-policy source in Section 5.

**Policy gradient RL algorithms.** For baselines that do not train on $\mathcal{D}_{\text{off}}$, we use REINFORCE (Ahmadian et al., 2024) (see Appendix F.1 for the off-policy RL baseline on $\mathcal{D}_{\text{off}}$). Starting from $\pi^0$, REINFORCE iteratively updates $\pi^t$ by ascending the return $J(\pi)$ using the gradient: $\mathbb{E}_{\boldsymbol{x} \sim \rho} \mathbb{E}_{\boldsymbol{y} \sim \pi^t(\cdot | \boldsymbol{x})} \left[ A_{\pi^t}(\boldsymbol{x}, \boldsymbol{y}) \cdot \log \pi^t(\boldsymbol{y} | \boldsymbol{x}) \right]$. Following (Guo et al., 2025), we estimate the expectation with $n$ sampled traces and use the group baseline $\hat{A}(\boldsymbol{x}, \boldsymbol{y}_i) = r(\boldsymbol{x}, \boldsymbol{y}_i) - 1/n \sum_{j=1}^{n} r(\boldsymbol{x}, \boldsymbol{y}_j)$. On hard problems with pass@$n \approx 0$, all $n$ samples typically fail, so $\hat{A}$ (and thus the gradient) is near zero, yielding the *stalling regime*.

# 3. PrefixRL:
# On-Policy RL on Very Off-policy Prefixes

In this section, we introduce the **PrefixRL** framework, which conditions on prefixes from $\mathcal{D}_{\text{off}}$ to guide on-policy exploration, boosting success on prefixed problems and transferring to the original no-prefix problems. We also prove some theoretical properties of the PrefixRL trained model. Before that, we discuss a more direct approach first.

**Training directly on $\mathcal{D}_{\text{off}}$.** Naïvely using the off-policy trace set $\mathcal{D}_{\text{off}}$ can hurt RL. *First*, SFT warm-starting on $\mathcal{D}_{\text{off}}$ often collapses entropy, so subsequent RL mostly sharpens a narrower response distribution and can reduce pass@$k$ even when pass@$1$ improves. Second, importance-weighted off-policy RL on $\mathcal{D}_{\text{off}}$ can be unstable due to large distribution shift and high-variance or biased gradients from practical weighting/clipping choices. See Figure 13 in Appendix C for the empirical evidence and further discussion.

**Creating prefixed problems in PrefixRL.** Instead of imitating off-policy traces, PrefixRL runs on-policy RL conditioned on off-policy prefixes (in addition to the original no-prefix problems). Crucially, the gradients are always masked on the off-policy prefix, avoiding the instability of policy gradients on very off-policy tokens. We create a dataset of prefixed problems $\mathcal{D}_{\text{pre}}$ by taking prefixes from a correct trace $\boldsymbol{y}^{\boldsymbol{x}} \in \mathcal{D}_{\text{off}}$ and appending the first $h$ tokens $(\boldsymbol{y}_{1:h}^{\boldsymbol{x}})$ in $\boldsymbol{y}^{\boldsymbol{x}}$ to the original problem $\boldsymbol{x}$, creating the prefixed problem $\text{concat}(\boldsymbol{x}, \boldsymbol{y}_{1:h}^{\boldsymbol{x}})$. We create multiple prefixed problems for every original problem by choosing multiple $h$. We choose prefix lengths $h$ such that, conditioned on the prefix, the base LLM has a reasonable accuracy under base LLM (see Section 5). Typically these are states revealing a high-level problem-solving strategy which the base LLM has little probability of sampling on its own (see Figure 3).

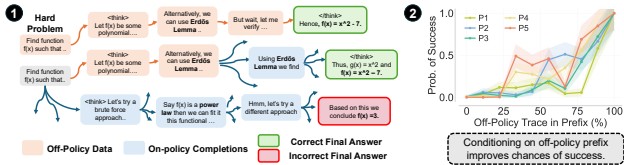

*Figure 3.* **Prefixing on off-policy trace improves probability of future success.** When we condition on prefixed problems, we increase the accuracy by placing the policy at key strategy-revealing states (Erdös Lemma in example ❶). For five problems (P1-P5) we plot accuracy when conditioning on prefixes of varying lengths, as a proportion of the full off-policy prefix length (❷).

**PrefixRL training objective.** The PrefixRL objective in (3.1) optimizes rewards on both prefixed problems in $\mathcal{D}_{\text{pre}}$ and no-prefix ones in $\mathcal{D}$ within a maximum context length of $H$ tokens. Note that the reward $r(\boldsymbol{x}, \cdot)$ for any prefixed problem $\boldsymbol{x}_{\text{pre}}$ is identical to its no-prefix counterpart.

$$\underbrace{\sum_{\boldsymbol{x}_{\text{pre}} \in \mathcal{D}_{\text{pre}}} \mathbb{E}_{\boldsymbol{y} \sim \pi(\cdot | \boldsymbol{x}_{\text{pre}})} \left[ r(\boldsymbol{x}_{\text{pre}}, \boldsymbol{y}) \right]}_{\text{Prefixed-Problems}} + J(\pi) \qquad (3.1)$$

## 3.1. PrefixRL Objective-Consistency and Sample-Efficiency Guarantees

Prefixing problems with partial correct traces should place the policy in a higher-rewarding state (Figure 3). The central theoretical question, however, is not whether learning the prefixed-problems are easier, but instead whether optimizing over such *augmented* problem sets provably improves the performance on the *original* RL objective $J(\pi)$.

In general, training on an altered input distribution could change the objective away from maximizing $J(\pi)$. We show that this is *not* the case for PrefixRL as long as the prefixes come from correct traces generated by a realizable policy. Concretely, we prove: (i) *objective consistency:* every maximizer of the PrefixRL objective is also a maximizer of $J(\pi)$; and (ii) *sample complexity guarantees and improvement over online RL:* for a natural policy gradient variant, PrefixRL achieves a smaller suboptimality bound, which translates to a smaller number of on-policy samples required to reach a given reward $J(\pi)$. In other words, we formally show that **PrefixRL reuses your FLOPs**: it converts information already paid for in logged prefixes into sample-complexity advantages over standard RL.

**PrefixRL objective is consistent with standard RL.** We make the following assumption that the prefixes are taken from the correct traces generated by a realizable policy.

**Assumption 3.1** (**Realizability and correctness of off-policy traces**). Assume that for any $(\boldsymbol{x}, \boldsymbol{y}) \in \mathcal{D}_{\text{off}}$: (i) the trace is correct: $r(\boldsymbol{x}, \boldsymbol{y}) = 1$, and (ii) the trace is realizable by $\mu$ if there exists an optimal policy $\mu \in \Pi$ s.t. $\mu(\boldsymbol{y} | \boldsymbol{x}) = 1$.

The next theorem states that as long as prefixes are taken

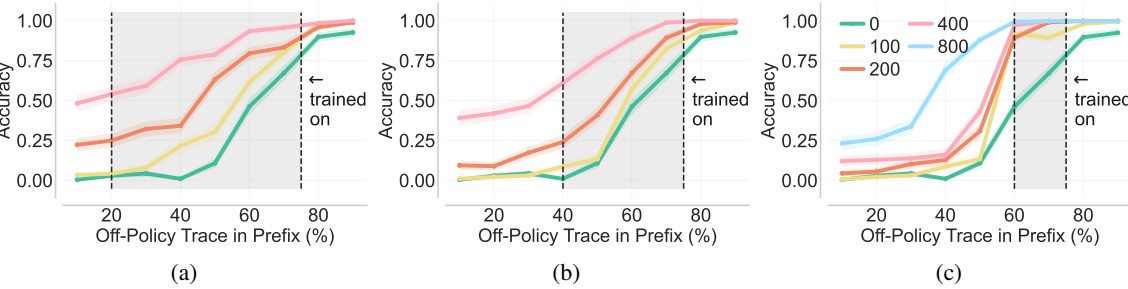

*Figure 4.* **Back-generalization (train/test mismatch):** We run RL only on prefixed-problems with prefix length in the shaded interval. We evaluate different training step checkpoints across the full range of prefix lengths, including no-prefix problems. Training on longer prefixes improves performance on shorter prefixes and can eventually improve no-prefix, indicating back-generalization (a,b). When training uses only very long prefixes (severe train/test mismatch), back-generalization to no-prefix problems takes more training steps (c).

from the correct traces realized by a policy $\in \Pi$, optimizing the PrefixRL objective preserves optimality on $J(\pi)$.

**Theorem 3.2** (**Consistency of the PrefixRL objective**). *Assume the realizability and correctness of off-policy traces (Assm. 3.1). Then, the maximizer of the PrefixRL objective (3.1) also maximizes standard RL objective $J(\pi)$.*

Intuitively, a maximizer of the PrefixRL objective produces correct traces on both no-prefix problems and prefixed problems. Since the prefixes come from correct traces, a good policy should also be able to complete the prefix to get the same reward; thus the two terms in the objective do not conflict with each other. Note that while PrefixRL does not change the global solution, it does not produce the same gradients as the standard RL objective.

**PrefixRL is more sample-efficient than standard RL.** Having established that PrefixRL does not bias policy optimization, we now quantify the benefits of prefixing in terms of the number of on-policy samples needed to reach a near-optimal policy. We analyze PrefixRL by instantiating the policy update to be natural policy gradient (Kakade, 2001) (PrefixRL-NPG). See Algorithm 1 and discussion in Appendix D.2. Concretely, each RL iteration alternates between: (i) *policy evaluation* by fitting critic or $Q$-function $\in \mathcal{F}$ using $N$ on-policy completions conditioned on the off-policy prefix, and (ii) *policy improvement* using the fitted critic via a mirror-descent update. In practice, algorithms like PPO, GRPO approximate the NPG update (Schulman et al., 2017). Theorem 3.3 bounds the suboptimality of the policy $\bar{\pi}_T$ returned by PrefixRL-NPG (Algorithm 1) in terms of the number of policy updates $T$, on-policy completions $N$, and a single distribution shift quantity between base policy $\pi_0$ and policy $\mu$ that realizes $\mathcal{D}_{\text{off}}$.

**Theorem 3.3** (**Suboptimality gap of PrefixRL**). *Under Assumption 3.1, let $\mathcal{D}_{\text{off}}$ be realized by $\mu \in \Pi$. For any $\delta \in [0, 1]$, with probability at least $1 - \delta$, policy $\bar{\pi}_T$ returned by PrefixRL-NPG (Algorithm 1) satisfies:* $\max_{\pi \in \Pi} J(\pi) - J(\bar{\pi}_T) \leq \mathcal{O}\left(\sqrt{\text{KL}(\mu \| \pi_0)/T} + \sqrt{1/N \cdot \log(T|\mathcal{F}|/\delta)}\right)$.

The bound decomposes into (i) an *optimization* term giving a $1/\sqrt{T}$ convergence rate (with an oracle evaluator, $N \to \infty$),

whose constant depends only on the initial shift between the base policy $\pi_0$ and the behavior policy $\mu$ realizing $\mathcal{D}_{\text{off}}$ and does not accumulate over iterations; in particular, if $\mathcal{D}_{\text{off}}$ comes from rejection sampling $\pi_0$ with at most $R$ attempts per problem, then $\mu = \pi_0$ conditioned on success and $\text{KL}(\mu \| \pi_0) = \mathcal{O}(\log R)$ so this dependence grows only logarithmically in $R$; and (ii) a *critic approximation* term from fitting $Q_{\pi_t} \in \mathcal{F}$ using $N$ on-policy traces, which in PrefixRL does not pay an extra distribution-shift penalty because sampling and evaluation use the same reset distribution induced by prefixes from $\mathcal{D}_{\text{off}}$ (unlike standard RL, which still suffers mismatch relative to $\pi^\star$). Moreover, Proposition 3.4 shows a reward function and $\pi^0$ where on-policy NPG and PrefixRL-NPG separate sharply (Appendix D.3): NPG is bottlenecked by exponentially rare rewarding traces, whereas PrefixRL-NPG needs only polynomial (in $H$) samples since much of the exploration is outsourced to the off-policy prefix.

**Proposition 3.4** (Worst-case separation with standard RL). *Let $\bar{\pi}_T^{\text{pre}}$ and $\bar{\pi}_T^{\text{std}}$ be the policies after $T$ iterations of Prefix and standard RL (states in Algorithm 1 $\sim \pi_t$). Then, there exists a reward function $r$ and base LLM $\pi^0$ such that $J(\bar{\pi}_T^{\text{pre}}) - J(\bar{\pi}_T^{\text{std}}) \geq 1 - \left(TN \cdot e^{-H}\right)$ for $TN = o(e^H)$.*

## 4. Back-Generalization Boosts the Learning Signal in PrefixRL

In Section 3.1, we showed that PrefixRL is consistent with standard RL except more sample-efficient. Now, we show that an empirical phenomenon we call **back-generalization** is a strong source of the gains behind PrefixRL and is unexplained by our theory. Back-generalization is defined as the performance improvement on no-prefix problems when we train *only* on their prefixed counterparts.

### 4.1. PrefixRL Improves No-Prefix Performance Even When Training Only on Prefixed-Problems

We run on-policy RL only on prefixed-problems where the prefix lengths are distributed uniformly between a fixed band of token-length percentiles of the full off-policy trace, but we evaluate accuracy across the full spectrum of prefix

lengths, including the no-prefix endpoint (0% prefixing). In Figure 4, we see generalization to no-prefix problems despite not having trained on them. This transfer from pre-fixed to no-prefix problems is particularly notable since the prefixes are highly unlikely under the base policy. When the training mixture includes relatively short prefixes, the mismatch is moderate (Figure 4 (a,b)). In this case, performance increases first near the trained band and then progressively improves for shorter prefixes, eventually lifting no-prefix accuracy. When training is restricted to very long prefixes (Figure 4(c)), the train/test mismatch with no-prefix problems is severe and the transfer is slower, but longer training (*e.g.*, 800 steps) still yields measurable no-prefix gains.

## 4.2. PrefixRL can Discover New Strategies Beyond What is Present in the Prefixed Problem

Clearly, back-generalization improves performance on unseen shorter prefixes, but the mechanism behind this is unclear. To understand this better, we create a simplified setup:

**Setup.** We run PrefixRL on the prefixes of a single off-policy trace in $\mathcal{D}_{\text{off}}$ (prefixed on >40% of the tokens in the trace). For two hard problems, we use a keyword heuristic to extract a salient "strategy" present in the off-policy trace for each problem. Then we track the prevalence of this keyword in (i) the prefixed-problem (PP) itself, (ii) the model's response when conditioned on that prefixed-problem (response for PP), and (iii) the early part of the trace on the original problem (untrained states in the model's response).

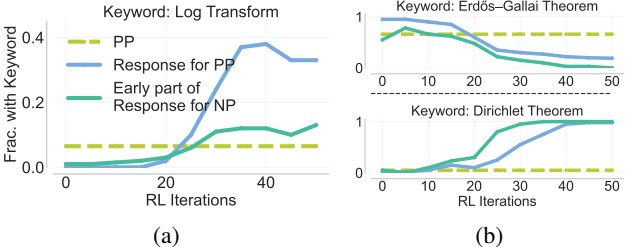

(a)                   (b)

*Figure 5.* **Strong coupling between responses for prefixed and no-prefix problems**: We train only on prefixed-problems (PP) and track the frequency of a strategy-indicating keyword in: (i) PP, (ii) model's response to PP, and (iii) the early part (prefixes not trained on) of the response to the no-prefix (NP) problem. There is a tight coupling of the strategies present in the responses for PP and NP (throughout RL), yet not purely imitative of the strategy explicitly provided in the prefixed-problem itself: the model can learn new strategies or suppress prefixed ones (*e.g.*, Erdős–Gallai).

Figure 5 probes how strategy usage evolves when running PrefixRL. The prefix is sampled from a fixed pool, so frequency in PP is constant (dashed line). In contrast, the response patterns change over training and reveal:

**Strategy usage is strongly correlated between prefixed and no-prefix responses.** There is a tight coupling between strategy use on prefixed and no-prefix problems, which is difficult to explain since no-prefix problems are ever trained on and many prefixed ones (>90% in (a)) may not even

contain the keyword. This suggests that PrefixRL updates internal state representations shared across responses for the prefixed and no-prefix problems. This further helps show that back-generalization indeed arises from favorable function approximation in LLMs and is not simply because solutions learned from training on the prefixed problem can be "stitched" into the no-prefix version, as hypothesized by prior work that prefixes on hints (Zhang et al., 2025b).

**PrefixRL can *unlearn* strategies in the prefixed-problem and discover new ones.** In Figure 5(b), we see that the policy at initialization uses the "Erdős–Gallai theorem" close to 90% of the time on the prefixed-problems since >50% of the prefixed-problems contain references to it. Throughout training, the frequency of traces mentioning "Erdős–Gallai" decreases steadily on responses to PP, indicating that PrefixRL can downweight suboptimal strategies in the off-policy prefixes. In Figure 5(b), we note that despite being conditioned on "Erdős–Gallai", RL upweights the rare (<2%) strategy at initialization ("Dirichlet Theorem"). This reweighting also transfers to the model's behavior on the no-prefix problem.

## 4.3. Which Prefixes Back-Generalize the Most in PrefixRL? Analysis via In-Context Learning

To study when back-generalization is effective, we analyze it in the *in-context learning* setting, where we run RL on problems prefixed with another problem and reasoning trace in context. This allows us to cleanly ablate the relationship between the off-policy prefix and the generated on-policy suffix based on how related the in-context problems are.

**Setup.** We run RL on a given problem with an entirely different problem (and its solution trace) in its context or prefix. Consider two problem sets: (P1, P3) where P1 and P3 are unrelated sub-problems, and (P2, P3) where P2 and P3 are related and solved with the same high level strategy (see Appendix E for details on P1, P2 and P3). We choose problems that are hard for the base model, with <1% pass@32.

**Back-generalization occurs when the prefix and suffix are sufficiently related.** From Figure 7, when the problems are related (P2 and P3), PrefixRL on P2 given P3 achieves 63% pass@4 on P2 and 60% pass@4 on the untrained in-context problem P3. Running standard RL on P2 alone predictably improves the pass@4 of P2 to 18% but the performance transfer to P3 is limited. In contrast, PrefixRL on unrelated problems (P3 | P1 or P1 | P2) performs similarly to doing RL on just P3 and P1 respectively. This suggests that *back-generalization is more effective when the components of the prefixed problem are related*, and in the in-context learning setting, back-generalization can be stronger than standard generalization across the two related problems. This setting could also shed light on a back-generalization mechanism for the standard reasoning setting, where LLMs often make $k$ attempts at the problem in a single trace. Since

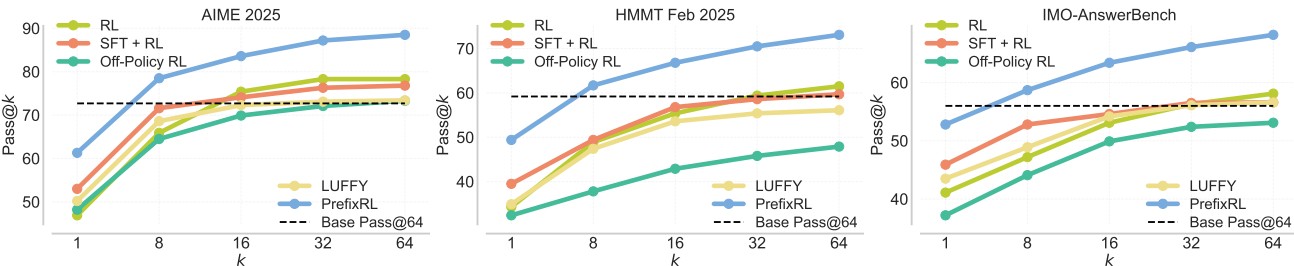

*Figure 6.* **Pass@$k$ on standardized evals**: We plot pass@k on AIME'25, HMMT'25 and IMO-AnswerBench for base LLM Llama-3.1-8B trained with PrefixRL, on-policy RL, off-policy RL, LUFFY and mark the base LLM's pass@64. All methods are run with $n = 8$.

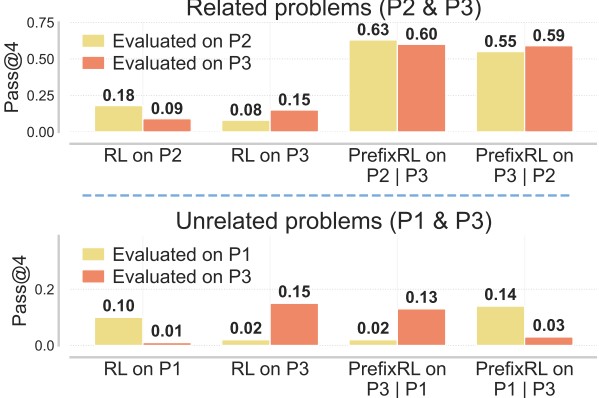

*Figure 7.* **Performance transfer via back-generalization can be stronger than typical generalization in RL:** When we prefix on the problem and full solution trace of one (in-context) problem (P2), and run PrefixRL to solve a different but related problem P3 | P2, we are able to improve performance on both P2 and P3 individually, and the transfer is much stronger compared to running RL on either problem individually. We do not see these gains when the in-context problem is unrelated in the case of P1 and P3.

the internal representations can be shared across related in-context problems so as to enable back-generalization, this could also be the case across attempts at a problem.

## 5. Experiments and Results

**Experimental setup.** We experiment with two thinking models: Llama-3.1-8B-instruct and Qwen3-4B-instruct. Since Llama-3.1-8B-instruct is not a thinking model, we distill it on OpenThoughtsV3 before all experiments, but still refer to it as Llama-3.1-8B (Guha et al., 2025). For training, we select 1k hard problems from DAPO (Yu et al., 2025) and OMNI-MATH (levels 6-8) (Gao et al., 2024), where pass@512 of Llama-3.1-8B is zero. We compare against on-policy RL (Ahmadian et al., 2024) and off-policy baselines that use $\mathcal{D}_{\text{off}}$: SFT (mid-training) on $\mathcal{D}_{\text{off}}$ followed by standard RL (SFT+RL), importance-weighted off-policy RL (Mahmood et al., 2014), and LUFFY (Yan et al., 2025). All evaluation results, unless noted, are on no-prefix problems. Unless otherwise stated, all main PrefixRL and RL experiments use REINFORCE with a group baseline. See

Appendix F for implementation details.

**Off-policy Dataset $\mathcal{D}_{\text{off}}$ and prefixed-problems.** For each base LLM, we produce $\mathcal{D}_{\text{off}}$ using large-scale rejection sampling until there is one correct off-policy trace per problem. We sample three prefixes of each off-policy trace at a uniformly random cut point between 40% and 80% of the tokens. The 3k prefixed and 1k no-prefix problems constitute the training data for PrefixRL.

**Compute accounting for off-policy data.** All compute-matched plots include the upfront rejection-sampling cost for PrefixRL, shown as the "off-policy compute" region. For the Llama3.1-8B experiments, this cost is $0.7 \times 10^{20}$ FLOPs, corresponding to about 650 sampled traces per problem on average across 1000 hard problems. Sampling difficulty varies substantially: some problems require about 150 samples to obtain a correct trace, while others require over 2000; we cap sampling at 2500. Overall, this gives roughly 650k sampled traces. By comparison, PrefixRL training uses over 400 iterations with 8 rollouts per problem and batch size 128; since training costs roughly $4\times$ sampling, this is at least $400 \times 8 \times 128 \times 4 \approx 1.64$M trace-equivalent samples.

**PrefixRL is $2\times$ more compute-efficient and achieves higher training accuracy.** Figure 2(left) and Figure 8(a) show that with the same compute, PrefixRL achieves higher accuracy on no-prefix problems compared to baselines for Llama-3.1-8B ($45\%$ greater) and Qwen-3-4B ($30\%$ greater) respectively. Even after accounting for the initial rejection-sampling cost, PrefixRL improves compute-efficiency by roughly $2\times$ over the strongest baseline (SFT+RL). In contrast, standard RL and SFT+RL only slowly improve accuracy even when the number of samples per problem $n$ is increased from 8 to 64. Thus, PrefixRL effectively re-allocates wasted sampling FLOPs in standard RL towards productively improving training rewards. Our iteration-matched plots (Figure 8(d)) show that our baselines have stable training curves, and increasing samples per problem $n$ unsurprisingly attains higher accuracy. Thus, PrefixRL's gains are not explained by degenerate baseline runs.

**PrefixRL improves both pass@1 and pass@$k$ on held-out**

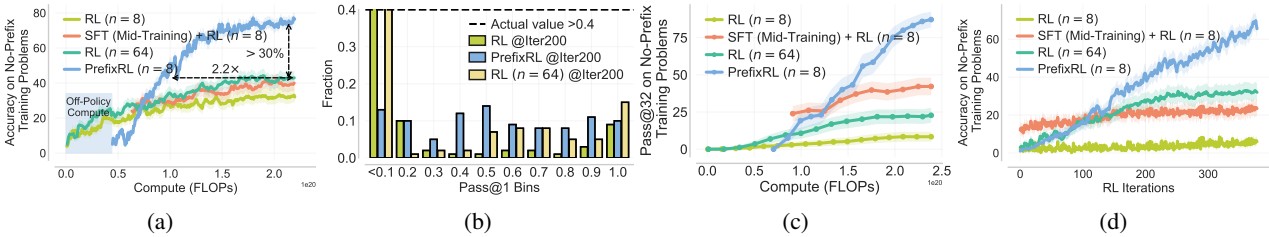

*Figure 8.* (**a**) **Qwen3-4b-instruct:** Compute-matched accuracy for PrefixRL and baselines with Qwen3-4B, using $\mathcal{D}_{\text{off}}$ from rejection sampling the base model. Results mirror the Llama3.1-8B setting (Figure 2); RL with $n = 64$ slightly beats SFT+RL, while PrefixRL remains $> 2\times$ more compute-efficient. (**b**) **Uniform pass@1 improvement:** By design, the base LLM places all training problems in the pass@1 bin $<0.1$. After 200 iterations, PrefixRL yields the most uniform gains across problems, while RL concentrates improvements on a small subset with rare successes; increasing $n$ partially mitigates this. (**c**) **New problems solved:** compute-matched pass@32 plots indicate that PrefixRL steadily expands the set of solvable problems rather than merely converting a fixed pass@$k$ (for small $k$) into higher pass@1, whereas the baselines largely saturate on pass@32. (**d**) **Iteration matched comparison:** iteration-matched reward curves confirm stable training across methods, so the compute-matched gains are not explained by unstable or degenerate baseline runs.

**benchmarks.** Figure 6 shows that on AIME'25, HMMT'25, and IMO-AnswerBench, PrefixRL improves pass@k for $k! \leq !64$ by at least 10% absolute over all baselines, including importance-weighting (off-policy RL) and LUFFY. This is notable as we only train on hard problems, so PrefixRL improves generalization to both easy and hard test problems. On AIME'25 with Llama-3.1-8B, PrefixRL raises pass@1 from 38.2 to 61.3; on HMMT'25, from 29.2 to 49.4. As $k$ increases, the gap widens (e.g., on AIME'25: $+18$ at $k=8$ and $+28$ at $k=64$), indicating that additional samples explore more promising subspaces rather than repeating low-value traces. Overall, PrefixRL improves both the mean performance (at $k=1$) and the tail (as $k$ grows).

**PrefixRL expands the set of solvable problems.** In Figure 8(c), PrefixRL steadily improves pass@32 (in addition to pass@1) while the baselines largely saturate in pass@32. This suggests that the baselines largely sharpen the distribution on already-solvable problems, while PrefixRL increases the set of solvable problems over RL training. Similarly, the pass@1 histogram in Figure 8(b) shows that standard RL concentrates improvements on a small subset of problems, whereas PrefixRL improves pass@1 uniformly across problems. Notably, non-uniform progress is known to cause training plateaus via *ray interference* (Schaul et al., 2019), which our findings also support.

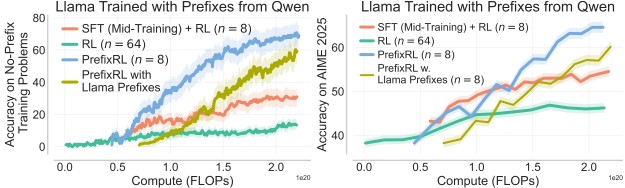

*Figure 9.* **PrefixRL remains effective with off-policy prefixes from a different model family.** We train Llama3.1-8B-Instruct using prefixed problems built from Qwen3-4B-Instruct rejection-sampled prefixes (left). Despite being more out of distribution, these prefixes improve hard-problem performance about as well as Llama-sourced prefixes (olive line). We also report AIME results for Llama trained with Qwen prefixes (right).

**PrefixRL remains effective with off-policy prefixes from a different model family.** When off-policy prefixes for PrefixRL come from Qwen3-4B-Instruct while the policy is based on Llama3.1-8B-Instruct (Figure 9), there is a similar train and test performance gain to PrefixRL with Llama-based prefixes. In fact, the Qwen prefixes outperform Llama prefixes in the compute-matched setting, despite being more off-policy. Note that this only holds in the compute-matched setting, since Qwen required less rejection sampling compute to collect one correct trace per problem. In the iteration-matched setting, Qwen and Llama-based prefixes perform similarly. The reverse, where we train Qwen with Llama prefixes, is also effective (Appendix F.3).

**Comparison to hint-based and off-policy baselines.** We compare PrefixRL to BREAD (Zhang et al., 2025b) and CHORD (Zhang et al., 2026), two recent methods for leveraging off-policy reasoning data during RL. For BREAD, we use the same correct off-policy traces, training problems, and RL config as PrefixRL, and instantiate its hinting procedure by treating the solution following the thinking trace as the expert hint. For CHORD, we use the same correct traces in its weighted SFT objective, with the annealed SFT loss and reward-shaping term. In compute-matched comparisons, PrefixRL outperforms both baselines on the original no-prefix hard training problems (Figure 10(left)).

This distinction is also reflected in back-generalization. Hint-based methods improve most when a large fraction of the reference solution is provided, but transfer less strongly back to the original no-hint problem (Figure 10(middle)). In contrast, PrefixRL trains on prefixed states that are treated as continuations of the model's own reasoning process, so improvements on prefixed problems more directly transfer to no-prefix problems (Figure 4). In a controlled single-problem experiment from Section 4.2, the off-policy trace uses an Erdős–Gallai-based solution, but PrefixRL learns to solve the original problem using a more robust Dirichlet-theorem strategy. BREAD, by contrast, remains more tied to the hinted strategy because the model is explicitly instructed

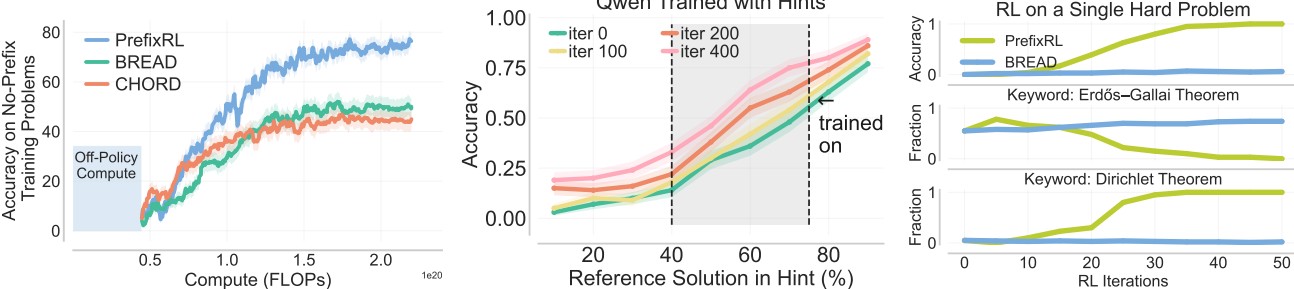

*Figure 10.* **PrefixRL outperforms hint-based and off-policy baselines. (Left)**: In a compute-matched comparison, PrefixRL outperforms hint-based approach BREAD (Zhang et al., 2025b) and off-policy approach CHORD (Zhang et al., 2026). **(Middle)**: Training with reference-solution hints yields weaker transfer back to no-hint problems compared to stronger back-generalization in PrefixRL (Figure 4). **(Right)**: On a single hard problem (setting from Section 4.2), PrefixRL is less constrained by the strategy in the off-policy trace (Erdős–Gallai) and learns a more robust strategy (Dirichlet-theorem), while BREAD remains tied to the provided hint strategy.

to use the provided reference hint. Thus, PrefixRL is closer to standard RL on the original task: the off-policy trace only moves the policy to a useful intermediate state, while the continuation is still optimized on-policy and remains free to explore alternative solution strategies. This finding agrees with our objective consistency result in Theorem 3.2.

### 5.1. Training Dynamics of PrefixRL are More Stable Compared to Typical Off-Policy Methods

In this section, we compare the RL training dynamics (like gradient variance) and the signal-to-noise ratio of policy gradients observed by PrefixRL and baselines, highlighting how PrefixRL sidesteps off-policy training instabilities.

**PrefixRL leverages off-policy data while preserving entropy for RL exploration.** Figure 11 (left) shows the average token-level entropy of the model's next-token distributions during the RL run. Notably, SFT on off-policy data causes a dramatic drop in entropy during RL, which could hurt RL exploration. In contrast, PrefixRL preserves the token-level entropy while still leveraging off-policy data.

**Fewer all-negatives on hard problems.** Figure 11 (middle) plots the fraction of *all-negative* problems (*i.e.*, where all $n$ samples get zero reward), measured *only* over no-prefix problems. PrefixRL consistently has a lower all-negative ratio than on-policy RL, revealing an underlying shift in the unconditional policy: the PrefixRL model is more likely to *enter* states where non-zero advantages are attainable (either due to the prefix revealing "useful" strategies that are further reinforced with positive rewards, or revealing likely but incorrect strategies that are useful to unlearn (Section 4.2)), thereby breaking the stalling regime (Section 2).

**PrefixRL achieves better accuracies with fewer sampled tokens.** Figure 11 (right) tracks the average number of sampled tokens in the response per problem. On the no-prefix problems, PrefixRL has shorter response lengths while achieving higher reward rates, implying better scaling with response length. Qualitatively, once the model

internalizes the strategy, it reaches decisive steps earlier, which reduces "unproductive wandering" later in the horizon. Moreover, since PrefixRL trains on a 3:1 mixture of prefixed to no-prefix problems and completing prefixed problems requires fewer tokens, the average number of tokens sampled (across all problem types) per batch is less than $1/2$ of the RL run (Figure 11 right, blue line). Since we do RL on shorter traces, the gradient variance for PrefixRL is much lower compared to standard RL (Figure 12 right).

**PrefixRL has higher signal-to-noise ratio during RL training.** Figure 12 shows the gradient norm and standard deviation of three methods during training: PrefixRL, importance weighted off-policy RL, and standard RL (see Appendix F.5 for details on gradient metrics). Following prior work (Liu et al., 2025a; Tan et al., 2025), we do weight-clipping and a biased token-level importance weighting correction for off-policy RL (Appendix F.1). Both off-policy RL and standard RL have low gradient norms. Since the rejection-sampled off-policy traces are unlikely under the policy, the token-level importance weight is very small (1e-3), which reduces the gradient norm for off-policy RL. PrefixRL has higher gradient norms since the off-policy prefixes are likely to place the policy in states with non-zero advantage (Figure 11 middle). Since PrefixRL samples fewer tokens during training (Figure 11 right), it achieves lower gradient variance as well (Figure 12 right). Overall, PrefixRL produces gradients with a higher norm and lower variance, accelerating RL with a higher signal-to-noise ratio.

## 6. Related Work and Discussion

**Using off-policy rollouts.** A standard fix for stalled on-policy search is to train on human or oracle traces (Lightman et al., 2023; Corrado et al., 2024), but teacher-driven methods inherit teacher limits (Agarwal et al., 2024) and often require reward shaping (Yan et al., 2025), entropy control (Wang et al., 2025a), and heavy tuning (Zhang et al., 2025a); on hard problems, long compatible traces are scarce

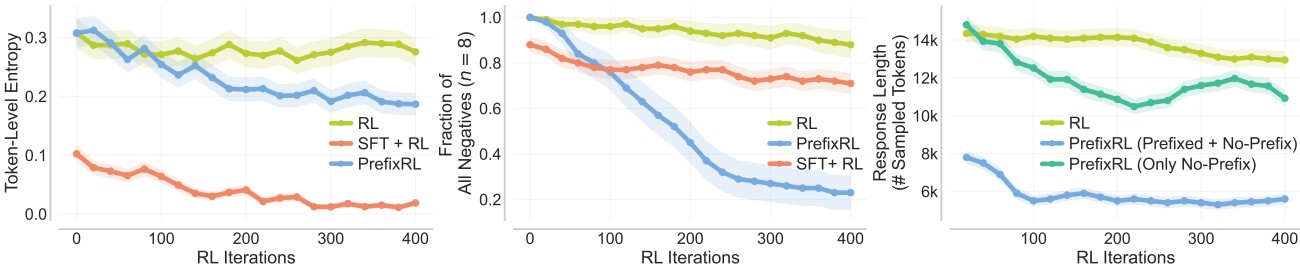

*Figure 11.* **RL Training Dynamics.** (**Left**) PrefixRL preserves token-level entropy during RL, whereas SFT may hurt exploration by lowering token-level entropy. (**Middle**) PrefixRL has lower "all negative ratio", or the number of no-prefix prompts with all zero rewards during RL. (**Right**) PrefixRL generates shorter responses on no-prefix problems (green) than standard RL, a source of gradient variance reduction. The average length across all problems (including prefixed ones) is much shorter and is a source of compute efficiency gains.

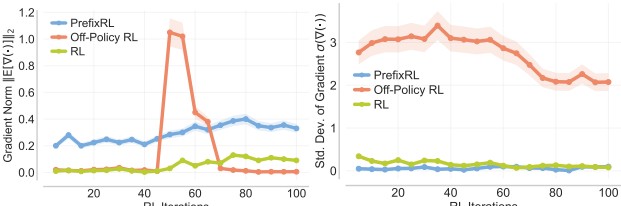

*Figure 12.* **PrefixRL has higher gradient signal-to-noise ratio.** PrefixRL simultaneously has higher gradient norm (left) and lower gradient variance (right) than standard RL and importance-weighted off-policy RL. For off-policy RL, importance weighting causes high gradient variance and a gradient norm spike.

and mismatch can reduce diversity (Kang et al., 2024b). Async reuse helps when behavior policies stay close in KL (Fu et al., 2025; Khatri et al., 2025), but large importance weights raise variance (Agarwal et al., 2021), so systems cap staleness (Sheng et al., 2024). Resetting to off-policy states is also classical (Kakade, 2003; Silver et al., 2016b; Agarwal et al., 2019; Daumé III et al., 2009). PrefixRL operationalizes these ideas for reasoning LLMs by conditioning on off-policy *prefixes* (not supervising on full trajectories), leveraging back-generalization (Sections 4) and improving compute allocation even after trace-collection cost.

**Exploration on hard problems.** RL can make small models outperform larger bases (Liu et al., 2025b; Luo et al., 2025) by reinforcing long-CoT behaviors (Qu et al., 2024; Gandhi et al., 2025), but can also shrink exploration and reduce pass@$k$ on hard instances (Yue et al., 2025; Zhao et al., 2025). Mitigations add exploration bonuses/regularizers or pass@$n$-aware objectives (Gao et al., 2025; Wang et al., 2025c; Song et al., 2025; Chow et al., 2024; Balashankar et al., 2025), or exploit asymmetries like the verification–generation gap and negative-gradient dynamics (Setlur et al., 2025b;a; Song et al., 2024; Zhu et al., 2025), yet models may still "under-think" (Wang et al., 2025b) and signal can come mainly from easier problems (He et al., 2024). Hint-based prompting guides RL with partial solutions (Chen et al., 2025b; Li et al., 2025; Qu et al., 2026; Amani et al., 2025) but typically assumes external traces and does not explain back-generalization (Zhang et al., 2025b). PrefixRL

instead reshapes the start-state distribution; uninformative prefixes recover standard on-policy RL (Section 4.3).

**Hint-based guidance.** PrefixRL is related to methods that guide RL with hints, including BREAD (Zhang et al., 2025b), AdaBack (Amani et al., 2025), QuestA (Li et al., 2025), POPE (Qu et al., 2026), and related methods (Chen et al., 2025b). PrefixRL instead reuses the base model's own prior sampling compute as prefix-conditioned start states, not supervision. Under compute-matched comparisons with the same off-policy traces, PrefixRL beats BREAD and exhibits stronger back-generalization. PrefixRL is also related to curriculum learning (Bengio et al., 2009; Narvekar et al., 2020; Florensa et al., 2017), since prefix lengths induce a spectrum of difficulty. We sample prefix lengths randomly rather than using an explicit curriculum; designing prefix curricula is a natural future direction.

## 7. Conclusion

PrefixRL is fundamentally different from typical methods that *imitate* off-policy data, relying instead on back-generalization to incorporate off-policy data as *conditioning* context while keeping updates on-policy. We expect the design space for such algorithms to be broad since conditioning is highly flexible and the mechanisms (and full potential) of back-generalization aren't fully understood. Crucially, back-generalization leverages the capabilities of the model itself to incorporate off-policy feedback. Thus, harnessing back-generalization is a crucial ingredient for bootstrapping prior compute and continuous self-improvement.

**Limitations.** Our main limitation is that the mechanism behind back-generalization is not yet fully understood and likely depends on how LLMs represent prefixes and intermediate reasoning states. Our experiments are also limited to math reasoning and mostly use base-model-generated traces, leaving open whether similar gains hold in domains such as code, planning, or agentic tool use. Future work should study broader off-policy sources, including scaffolded agent rollouts, traces from later RL checkpoints, and curriculum-based selection over prefix lengths.

## Impact Statement

This work evaluates methods for improving exploration and learning in reasoning-focused LLMs on structured, publicly available mathematical reasoning benchmarks, which are generally low-risk and unlikely to elicit toxic or sensitive content compared to open-ended generation. Nonetheless, advances that make models better at difficult problem solving can have dual-use potential if applied to malicious objectives (e.g., scalable deception or misinformation), so we view this progress as best paired with continued research on safety, alignment, and deployment safeguards, with authors retaining responsibility for all technical claims and conclusions.

## Acknowledgements

We would like to thank Sean Bell, Aviral Kumar, Rishabh Agarwal, Shengjia Zhao, Hongyu Ren, Yuanzhi Li, Ankur Pai, Yang Song, Saurabh Garg, Wen Sun, Max Simchowitz, Sergey Levine, Yuxiao Qu, Ian Wu, Rohan Maheshwari, and Yuandong Tian for helpful discussions and thoughtful feedback on our work. AS acknowledges generous support from the JP Morgan AI PhD fellowship, Laude Slingshot and the CMU Presidential fellowship.

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

# Appendices

## A. Full Set of Notations

**Markov decision process.** We use $\boldsymbol{x}$ to denote an input problem and $\boldsymbol{y} = (y_1, \ldots, y_H)$ for a response of $H$ tokens, and if $\boldsymbol{y} \sim \pi(\cdot \mid \boldsymbol{x})$, then $\boldsymbol{y}$ is sampled auto-regressively from the LLM $\pi$ fed with input $\boldsymbol{x}$. Each token in this response $\boldsymbol{y}$ belongs to a set of tokens or actions $\mathcal{A}$. The state $\boldsymbol{s}_h$ at time step $h$ is given by $(\boldsymbol{x}, y_1, y_2, \ldots, y_h)$, where the initial state $\boldsymbol{s}_0$ is just the problem $\boldsymbol{x}$. The set of states across all time steps is denoted by the class $\mathcal{S}$. We use $d_h^\pi$ to denote the distribution over states $\boldsymbol{s}_h$ at time step $h$ by rolling out the policy $\pi$ auto-regressively for $h$ time steps. For compactness, we write the trajectory-level log-likelihood $\log \pi(\boldsymbol{y} \mid \boldsymbol{x}) = \sum_{h=1}^{|\boldsymbol{y}|} \log \pi(y_h \mid \boldsymbol{x}, \boldsymbol{y}_{<h})$. For each problem we have access to outcome reward function $r(\boldsymbol{x}_i, \boldsymbol{y})$ to check whether the final answer in response $\boldsymbol{y}$ is correct/incorrect (1/0) for the problem $\boldsymbol{x}_i$ (*e.g.*, by matching the boxed answer in the end of $\boldsymbol{y}$ for math problems).

**Dataset of hard problems and off-policy traces.** We use $\mathcal{D}$ to denote a dataset of $N$ hard problems $\mathcal{D} =: \{\boldsymbol{x}_i\}_{i=1}^N$. We use $\pi^0$ to denote the base pre-trained LLM that we initialize the RL algorithm, $\pi^t$ as the policy after $t$ RL iterations and $\mathcal{D}_{\text{off}}$ as the dataset of off-policy traces. Finally, we define the pass rate @$k$ for problem $\boldsymbol{x}$ and LLM $\pi$ as $\mathbb{E}_{\boldsymbol{y}_1, \ldots, \boldsymbol{y}_k \sim \pi(\cdot \mid \boldsymbol{x})} \max(\{r(\boldsymbol{x}, \boldsymbol{y}_i)\}_{i=1}^k)$. In the main paper, we define the set of hard problems as those with pass@$k \approx 0$ under the base LLM $\pi^0$. See Section 5 for how we select these low pass rate hard problems for training.

## B. Additional Related Work

In our work, we present PrefixRL as a way to recycle old inference FLOPs in the form of very off-policy data for RL training of LLMs. In particular, we test the efficacy of this framework when we wish to train on hard low pass rate problems for which on-policy sampling rarely produces rollouts and exploration becomes a major bottleneck. Now, we briefly discuss a few lines of related work on off-policy RL, exploration on hard problems and a few others that improve on-policy exploration by conditioning the policy on tailored guidance and hints.

**Learning from off-policy LLM rollouts.** When on-policy search stalls due to over-sharpening or "over-thinking," a common approach is to supervise on human or oracle-provided traces (Lightman et al., 2023; Corrado et al., 2024), but teacher-driven methods inherit the teacher's capacity limit (Agarwal et al., 2024) and often require reward shaping (Yan et al., 2025), entropy control (Wang et al., 2025a), and heavy hyperparameter tuning (Zhang et al., 2025a); moreover, for hard problems, long model-compatible chains of thought are scarce and mismatches can collapse response diversity (Kang et al., 2024b). When off-policy data come from "close enough" (in KL divergence) policies as in Async RL, reuse becomes more efficient (Fu et al., 2025; Khatri et al., 2025), yet large importance weights and high gradient variance pose instability risks (Agarwal et al., 2021), so practical systems cap behavior-policy staleness to only a few RL iterations (Sheng et al., 2024). These constraints motivate approaches that do not treat off-policy trajectories as direct supervision targets; related directions condition on subgoals or plans (Hong et al., 2025), higher-level abstractions (Qu et al., 2025), or partial solutions (Amani et al., 2025; Chen et al., 2025b; Li et al., 2025). Different from the above, PrefixRL conditions on off-policy prefixes from long-thinking traces, as opposed to training on them before or during RL. Instead of suffering from instability due to supervising on off-policy data (Sections 3 and 5.1), PrefixRL benefits from them via back-generalization.

**Conditioning on hints to improve on-policy RL.** A related line of work augments prompts with hints or partially revealed human solutions to "guide" on-policy RL (Chen et al., 2025b; Li et al., 2025; Qu et al., 2026). AdaBack (Amani et al., 2025) adaptively searches for the minimal hint that improves performance over human-written solutions, but is hard to scale to long-context "thinking models" and large datasets. Similarly, QuestA (Li et al., 2025) uses answer-hinted prompts derived from human solutions. In general, these methods are only feasible when we have access to solution traces written by a human or a more capable teacher model. In contrast, PrefixRL enables a self-improvement loop by not relying on external sources and instead reusing compute from prior models. Moreover, our work also analyzes the back-generalization phenomenon that may be shared across these methods, showing that it cannot be explained by some of the "stitching" arguments made in prior works (Zhang et al., 2025b).

**Resetting to off-policy states in RL.** The idea of "resetting" current RL policy to off-policy states is not new in RL (Kakade, 2003; Bagnell et al., 2003; Nair et al., 2018; Salimans & Chen, 2018; Yin et al., 2022; Uchendu et al., 2023; Silver et al., 2016a;b; Agarwal et al., 2019; Daumé III & Marcu, 2005; Daumé III et al., 2009). Our contribution is to instantiate this perspective for RL of reasoning LLMs. We show that a relatively small dataset of *correct* off-policy traces is sufficient to

enable effective resets that make hard, low-pass-rate problems trainable even when on-policy rollouts almost never succeed. We also show that PrefixRL yields a strictly better allocation of compute, even after accounting for the inference cost of collecting the off-policy traces.

**Improving exploration on hard problems in LLM reasoning.** Small models fine-tuned with RL can outperform much larger base models (Liu et al., 2025b; Luo et al., 2025), largely by reinforcing long chain-of-thought behaviors like self-correction (Qu et al., 2024) and reflection (Gandhi et al., 2025). Yet, without careful controls, RL often under-explores and leaves hard instances underprobed; empirically this appears as a drop in pass@$k$ versus the base model (Yue et al., 2025; Zhao et al., 2025). One response is to regularize training to curb over-sharpening via intrinsic-motivation bonuses (Gao et al., 2025), entropy (Wang et al., 2025c), count-based signals (Song et al., 2025), or objectives that directly optimize pass@$n$ (Chow et al., 2024; Balashankar et al., 2025), but these still inherit sparse-reward limits and depend on easy problems for signal (He et al., 2024). A complementary thread (Setlur et al., 2025b) exploits base-model asymmetries, *e.g.*, the verification–generation gap (Setlur et al., 2025a; Song et al., 2024), and can combine with negative-gradient dynamics to chain such asymmetries across updates (Zhu et al., 2025); nevertheless, models often "under-think" (Wang et al., 2025b), persisting with wrong high-level plans despite more rollouts. In contrast, PrefixRL avoids carefully tuned auxiliary exploration objectives by reshaping the start-state distribution directly. Empirically, we do not observe the pass@$k$ regressions often induced by over-sharpening or over exploration with token-level entropy regularizers. In the worst case, uninformative prefixes recover standard on-policy RL (Section 4.3).

## C. Using Off-Policy Traces as Supervision Targets

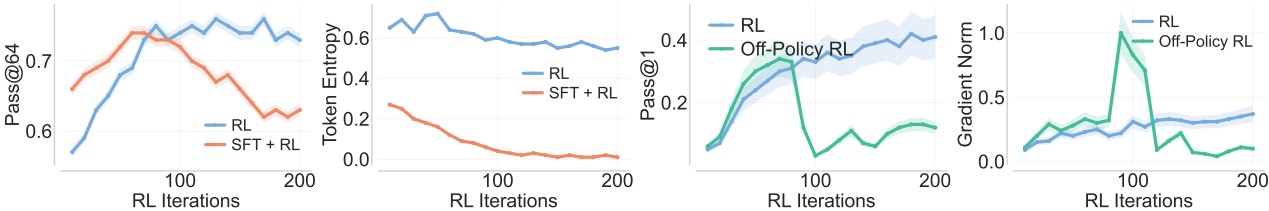

*Figure 13.* **Supervising the policy on $\mathcal{D}_{\text{off}}$ can cause diversity collapse or training instabilities during RL: (a, b)** Warm-starting the RL run by running SFT on $\mathcal{D}_{\text{off}}$ before (mid-training) reduces token entropy (a) and hurts exploration during RL (worse Pass@64 performance in (b)). **(c, d)** Directly using $\mathcal{D}_{\text{off}}$ during online RL by updating the current RL policy importance-weighted off-policy traces (in addition to on-policy traces) leads to training instabilities. We see the gradient norm (clipped at 1.0) blow up during training (d) and this leads to optimization collapse (c).

**SFT on $\mathcal{D}_{\text{off}}$ can hurt exploration during RL.** A common way to improve RL on hard problems is to first mid-train (SFT) on traces in $\mathcal{D}_{\text{off}}$ to warm-start RL training. However, while SFT boosts post-RL pass@1, pass@64 drops after about 100 iterations (Figure 13(a)). This loss of diversity is driven by a sharp entropy collapse after SFT, which decreases further during RL (Figure 13(b)). This suggests that here, RL mainly sharpens the distribution of responses the model can already produce after SFT. In contrast, an early-stopped SFT checkpoint can underfit and encourage random exploration during RL (Wang et al., 2025d). One can partially mitigate this by enlarging the SFT dataset ($\mathcal{D}_{\text{off}}$), but doing so increases the upfront cost of the SFT step.

**Off-policy RL using $\mathcal{D}_{\text{off}}$ leads to training instabilities.** A more direct way to use the off-policy data $\mathcal{D}_{\text{off}}$ is to do importance-weighted off-policy RL (Degris et al., 2012) (see Appendix F.1), which accounts for the distribution shift between the current RL and sampling (rejection sampling on base LLM) policies. However, this suffers from large gradient variance or heavily biased gradients due to clipping and token-level weighting rather than sequence-level (Agarwal et al., 2021) (Section 5.1). This can cause training reward collapse and unstable optimization (Figures 13(c),(d)) as we force updates on very unlikely token sequences under the current RL policy, leading to memorization of $\mathcal{D}_{\text{off}}$ (Kang et al., 2024a; Setlur et al., 2024).

## D. Omitted Proofs

In this section, we present proofs for our theoretical results in Section 3.1. We begin with the proof for Theorem 3.2 which implies that the PrefixRL objective is consistent with standard RL, and any solution for our PrefixRL objective, is also a mazimizer of the standard RL objective which just maximizes $J(\pi)$. Following this, we show the proof for Theorem 3.3

which bounds the suboptimality gap of an algorithm using natural policy gradient (NPG) to optimize the PrefixRL objective. Note that this is slightly different from the policy gradients we use in practice, but is nevertheless insightful in informing a formal mental model for the gains behind PrefixRL. We also provide a proof for Proposition 3.4 that lower bounds the performance gap between PrefixRL and standard RL in the worst case. In the end we list auxiliary lemmas useful for analysis.

### D.1. Proof of Theorem 3.2

Let the standard (no-prefix) RL objective be

$$J(\pi) \;=\; \sum_{\boldsymbol{x}\in\mathcal{D}} \mathbb{E}_{\boldsymbol{y}\sim\pi(\cdot|\boldsymbol{x})}\big[r(\boldsymbol{x},\boldsymbol{y})\big], \qquad J^{\star} \;=\; \max_{\pi\in\Pi} J(\pi). \tag{D.1}$$

For each $\boldsymbol{x}\in\mathcal{D}$, Assumption 3.1 gives a single correct trace $\boldsymbol{y}^{\boldsymbol{x}}$ such that $(\boldsymbol{x},\boldsymbol{y}^{\boldsymbol{x}})\in\mathcal{D}_{\text{off}}$. Given any cut index $h$, define the prefixed problem

$$\boldsymbol{x}_{\text{pre}} \;=\; \text{concat}(\boldsymbol{x},(\boldsymbol{y}^{\boldsymbol{x}})_{1:h}), \tag{D.2}$$

and define its reward by evaluating the full transcript:

$$r(\boldsymbol{x}_{\text{pre}},\boldsymbol{z}) \;:=\; r\big(\boldsymbol{x},(\boldsymbol{y}^{\boldsymbol{x}})_{1:h}\circ\boldsymbol{z}\big).$$

The PrefixRL objective is

$$J_{\text{pre}}(\pi) \;=\; \sum_{\boldsymbol{x}_{\text{pre}}\in\mathcal{D}_{\text{pre}}} \mathbb{E}_{\boldsymbol{z}\sim\pi(\cdot|\boldsymbol{x}_{\text{pre}})}\big[r(\boldsymbol{x}_{\text{pre}},\boldsymbol{z})\big] \;+\; \sum_{\boldsymbol{x}\in\mathcal{D}} \mathbb{E}_{\boldsymbol{y}\sim\pi(\cdot|\boldsymbol{x})}\big[r(\boldsymbol{x},\boldsymbol{y})\big]. \tag{D.3}$$

**A uniform upper bound.** Fix any $\boldsymbol{x}\in\mathcal{D}$ and any prefix $(\boldsymbol{y}^{\boldsymbol{x}})_{1:h}$ used to form a prefixed problem. For any policy $\pi\in\Pi$ defined on such prefixed problems, construct a policy $\tilde{\pi}\in\Pi$ for the no-prefix problem $\boldsymbol{x}$ that deterministically emits the prefix $(\boldsymbol{y}^{\boldsymbol{x}})_{1:h}$ and then samples the suffix from $\pi(\cdot\mid\text{concat}(\boldsymbol{x},(\boldsymbol{y}^{\boldsymbol{x}})_{1:h}))$. By the reward definition,

$$\mathbb{E}_{\boldsymbol{y}\sim\tilde{\pi}(\cdot|\boldsymbol{x})}\big[r(\boldsymbol{x},\boldsymbol{y})\big] \;=\; \mathbb{E}_{\boldsymbol{z}\sim\pi(\cdot|\text{concat}(\boldsymbol{x},(\boldsymbol{y}^{\boldsymbol{x}})_{1:h}))}\big[r(\text{concat}(\boldsymbol{x},(\boldsymbol{y}^{\boldsymbol{x}})_{1:h}),\boldsymbol{z})\big].$$

Therefore, for every $\pi$ and every such prefixed problem $\boldsymbol{x}_{\text{pre}}=\text{concat}(\boldsymbol{x},(\boldsymbol{y}^{\boldsymbol{x}})_{1:h})$,

$$\mathbb{E}_{\boldsymbol{z}\sim\pi(\cdot|\boldsymbol{x}_{\text{pre}})}\big[r(\boldsymbol{x}_{\text{pre}},\boldsymbol{z})\big] \;\leq\; \max_{\pi'\in\Pi}\mathbb{E}_{\boldsymbol{y}\sim\pi'(\cdot|\boldsymbol{x})}\big[r(\boldsymbol{x},\boldsymbol{y})\big]. \tag{D.4}$$

Let $m(\boldsymbol{x})$ be the number of prefixed problems in $\mathcal{D}_{\text{pre}}$ derived from $\boldsymbol{x}$. Summing the above inequality over all prefixed problems and grouping by their originating $\boldsymbol{x}$ gives

$$\sum_{\boldsymbol{x}_{\text{pre}}\in\mathcal{D}_{\text{pre}}} \mathbb{E}\big[r(\boldsymbol{x}_{\text{pre}},\cdot)\big] \;\leq\; \sum_{\boldsymbol{x}\in\mathcal{D}} m(\boldsymbol{x})\max_{\pi'\in\Pi}\mathbb{E}_{\boldsymbol{y}\sim\pi'(\cdot|\boldsymbol{x})}\big[r(\boldsymbol{x},\boldsymbol{y})\big]. \tag{D.5}$$

Define the constant

$$C \;:=\; \sum_{\boldsymbol{x}\in\mathcal{D}} m(\boldsymbol{x})\max_{\pi'\in\Pi}\mathbb{E}_{\boldsymbol{y}\sim\pi'(\cdot|\boldsymbol{x})}\big[r(\boldsymbol{x},\boldsymbol{y})\big], \tag{D.6}$$

which is independent of $\pi$. Then for every $\pi\in\Pi$ we have the uniform upper bound

$$J_{\text{pre}}(\pi) \;\leq\; C + J(\pi). \tag{D.7}$$

**Tightness using Assumption 3.1.** By realizability in Assumption 3.1, there exists a policy $\mu\in\Pi$ such that

$$\mu(\boldsymbol{y}^{\boldsymbol{x}}\mid\boldsymbol{x}) = 1, \qquad \forall\boldsymbol{x}\in\mathcal{D}. \tag{D.8}$$

In particular, for any cut $h$, conditioning $\mu$ on the prefix $(\boldsymbol{y}^{\boldsymbol{x}})_{1:h}$ yields the deterministic continuation:

$$\mu\big(\boldsymbol{z}\mid\text{concat}(\boldsymbol{x},(\boldsymbol{y}^{\boldsymbol{x}})_{1:h})\big) \;=\; \mathbf{1}\{\boldsymbol{z}=(\boldsymbol{y}^{\boldsymbol{x}})_{h+1:|\boldsymbol{y}^{\boldsymbol{x}}|}\}. \tag{D.9}$$

Therefore, on every prefixed problem $\boldsymbol{x}_{\mathrm{pre}} = \mathrm{concat}(\boldsymbol{x}, (\boldsymbol{y}^{\boldsymbol{x}})_{1:h})$,

$$\mathbb{E}_{\boldsymbol{z} \sim \mu(\cdot|\boldsymbol{x}_{\mathrm{pre}})}\big[r(\boldsymbol{x}_{\mathrm{pre}}, \boldsymbol{z})\big] \ = \ r(\boldsymbol{x}, \boldsymbol{y}^{\boldsymbol{x}}) \ = \ 1, \tag{D.10}$$

where the last equality uses correctness of $\mathcal{D}_{\mathrm{off}}$ in Assumption 3.1. Hence,

$$\sum_{\boldsymbol{x}_{\mathrm{pre}} \in \mathcal{D}_{\mathrm{pre}}} \mathbb{E}_{\boldsymbol{z} \sim \mu(\cdot|\boldsymbol{x}_{\mathrm{pre}})}\big[r(\boldsymbol{x}_{\mathrm{pre}}, \boldsymbol{z})\big] \ = \ \sum_{\boldsymbol{x} \in \mathcal{D}} m(\boldsymbol{x}). \tag{D.11}$$

Moreover, by (D.8) and correctness,

$$J(\mu) \ = \ \sum_{\boldsymbol{x} \in \mathcal{D}} \mathbb{E}_{\boldsymbol{y} \sim \mu(\cdot|\boldsymbol{x})}\big[r(\boldsymbol{x}, \boldsymbol{y})\big] \ = \ \sum_{\boldsymbol{x} \in \mathcal{D}} r(\boldsymbol{x}, \boldsymbol{y}^{\boldsymbol{x}}) \ = \ |\mathcal{D}|. \tag{D.12}$$

Therefore $J^{\star} = |\mathcal{D}|$ and $J(\mu) = J^{\star}$. Since rewards are in $[0, 1]$, for each $\boldsymbol{x}$ we have $\max_{\pi' \in \Pi} \mathbb{E}_{\boldsymbol{y} \sim \pi'(\cdot|\boldsymbol{x})}[r(\boldsymbol{x}, \boldsymbol{y})] = 1$, and thus the constant simplifies to $C = \sum_{\boldsymbol{x} \in \mathcal{D}} m(\boldsymbol{x})$. Combining with (D.11) yields

$$J_{\mathrm{pre}}(\mu) \ = \ C + J(\mu) \ = \ C + J^{\star}. \tag{D.13}$$

**Step 3: Concluding the consistency of PrefixRL objective.** Let $\hat{\pi} \in \arg\max_{\pi \in \Pi} J_{\mathrm{pre}}(\pi)$ be any maximizer of the PrefixRL objective. By optimality of $\hat{\pi}$, (D.13), and the upper bound (D.7),

$$C + J(\hat{\pi}) \ \geq \ J_{\mathrm{pre}}(\hat{\pi}) \ \geq \ J_{\mathrm{pre}}(\mu) \ = \ C + J^{\star}. \tag{D.14}$$

Cancelling $C$ yields $J(\hat{\pi}) \geq J^{\star}$, hence $J(\hat{\pi}) = J^{\star}$. Therefore $\hat{\pi} \in \arg\max_{\pi \in \Pi} J(\pi)$, proving that any maximizer of the PrefixRL objective also maximizes the standard RL objective.

### D.2. Proof of Theorem 3.3

In this section we present our proof for Theorem 3.3 which bounds the performance suboptimality for PrefixRL. In particaly, we bound thisgap for an algorithm that conforms to the PrefixRL workflow (Algorithm 1) but uses natural policy gradient (NPG) (Kakade, 2001) to update the policy iteratively (starting from the base LLM $\pi^0$). In our practical implementation use REINFORCE to compute on-policy gradients. Next we introduce the setup and describe the key steps in Algorithm 1, some differences with practice and the full proof.

**Setup.** We use $\pi^0$ to denote the base LLM we start RL training with, and $\mu$ as the policy used to IID sample the dataset of off-policy traces $\mathcal{D}_{\mathrm{off}}$, one trace for each problem in $\boldsymbol{x} \in \mathcal{D}$. We assume that $\mathcal{D}_{\mathrm{off}}$ is realizable (Assumption 3.1) which implies that it only consists of correct off-policy traces. Let $\mathcal{F}$ denote the class of $Q$-functions induced by all policies in the policy class $\Pi$, and $H$ be the maximum context length or horizon $H$ of the auto-regressive Markov decision process (MDP) induced by the policies in $\Pi$ and reward function $r$.

**Description of PrefixRL with NPG (Algorithm 1).** In each iteration of Algorithm 1 we *first* collect a dataset of state, action and reward triplets $\mathcal{D}_t$. *Second*, we fit a critic or $Q$ function $\hat{Q}^t$ on this dataset of size $N$ (step 9). *Third*, we use the fitted $Q$ function to perform a state-wise mirror ascent or natural policy update (step 11) in order to get the subsequent RL iterate. We collect the $N$ traces in $\mathcal{D}_t$ by uniformly sampling an off-policy state $s_h$ (prefixed problem) from $\mathcal{D}_{\mathrm{off}}$. Then, we rollout the current RL policy $\pi^t$ conditioned on state $s_h$ to sample a single action or token $a_h$ (step 6). To estimate the $Q$ function under the current RL policy at this state-action pair we now complete the rollout till time step $H$ and collect a reward (step 7).

**Difference with practice: Algorithm 1 uses $Q$ functions instead of direct rewards.** The update in NPG is similar to REINFORCE except that we use $N$ on-policy samples to first estimate a $Q$ function (step 10) in the $Q$ function class $\mathcal{F}$) for the current RL iterate $\pi^t$ and then use the estimated $Q$ function to update the policy and get $\pi_{t+1}$ using mirror ascent (step 12). This is a bit different from REINFORCE where we compute the policy gradient using only the rewards attain by the $N$ on-policy traces and perform a step of gradient.

**Difference with practice: Algorithm 1 samples new prefixed problems from $\mathcal{D}_{\mathrm{off}}$.** In practice we construct a prefixed problems from a fixed dataset of off-policy traces $\mathcal{D}_{\mathrm{off}}$ and also use the set of prefixed problems in $\mathcal{D}_{\mathrm{pre}}$ are fixed throughout RL training. In contrast, Algorithm 1 samples off-policy states (prefixed problems) from the dataset of off-policy traces $\mathcal{D}_{\mathrm{off}}$.

---

**Algorithm 1** PrefixRL with Natural Policy Gradients

---

**Require:** Base policy $\pi^0$, off-policy data $\mathcal{D}_{\text{off}}$, horizon $H$, iterations $T$, step size $\eta$, $Q$ function class $\mathcal{F}$

1: Initialize $\pi^1 \leftarrow \pi^0$
2: **for** $t = 1, \ldots, T$ **do**
3:     Initialize dataset $\mathcal{D}_t \leftarrow \emptyset$
4:     **for** $i = 1, \ldots, n$ **do**
5:         Sample $(\boldsymbol{s}_h, a_h^{\text{off}})$ uniformly from state-action pairs in $\mathcal{D}_{\text{off}}$ {sample prefixed-problem}
6:         Sample $a_h \leftarrow a_h^{\text{off}}$ with probability $\frac{1}{2}$, else sample $a_h \sim \pi^t(\cdot \mid \boldsymbol{s}_h)$
7:         Execute $\pi^t(\cdot \mid \boldsymbol{s}_h, a_h)$ from step $h+1$ through $H$ to obtain terminal reward $r$
8:         $\mathcal{D}_t \leftarrow \mathcal{D}_t \cup \{(\boldsymbol{s}_h, a_h, r)\}$
9:     **end for**
10:     **Critic fit (regression oracle):**
11:         $\hat{Q}^t \leftarrow \arg\min_{f \in \mathcal{F}} \sum_{(\boldsymbol{s}, a, r) \in \mathcal{D}_t} (f(\boldsymbol{s}, a) - r)^2$
12:     **Natural policy update (mirror ascent):** {performed state-wise}
13:         $\pi^{t+1}(\cdot \mid \boldsymbol{s}) \leftarrow \arg\min_p \langle -\hat{Q}^t(\boldsymbol{s}, \cdot), p \rangle + \frac{1}{\eta} \text{KL}(p \,\|\, \pi^t(\cdot \mid \boldsymbol{s}))$
14: **end for**
15: **return** $\bar{\pi}_T \leftarrow \frac{1}{T} \sum_{t=1}^T \pi^t$ {return mixture policy}

---

This difference is pretty minor but perhaps underscores the performance improvements driven by back-generalization in being able to improve performance on the original no-prefix problems despite PrefixRL only using a small fraction of all possible off-policy prefixes in $\mathcal{D}_{\text{off}}$.

**Comparison with Chang et al. (2024).** Our proof technique follows Chang et al. (2024), adapting to the setting of verifiable rewards with our different "reset" policy (which we refer to as prefix policy $\mu$), and removing the requirement of KL divergence between the current and the reset policy. Since our off-policy dataset consists of only realizable correct traces we will need much weaker assumptions. Following are some key differences compared to Algorithm 3 in (Chang et al., 2024) that allows us to prove the suboptimality gap with weaker assumptions. *First*, we sample the prefix from the comparator policy (in other words the prefix generating policy is realizable and lies in the class of optimal policies). This ensures sufficient coverage for the distribution of Q-function regression (ensuring small error in fitting the critic) over states visited by a "good" policy even though the current RL iterate is far from it. *Second*, we output the mixture policy (standard in self-play literature (Bai et al., 2020; Hofbauer & Sorin, 2006)). *Finally*, unlike Chang et al. (2024), we don't require a bound on the KL divergence against the SFT policy or the policy trained on the off-policy data.

**Assumptions needed for Theorem 3.3.** Now, we list the assumptions we make in our analysis of the suboptimality gap of PrefixRL. In general, they are milder than the assumptions in Chang et al. (2024).

- Assumption D.1 is pretty standard in the analysis of actor-critic methods (Haarnoja et al., 2018) and only requires that our critic function class is expressive enough to realize the $Q$ function induced by any policy in $\Pi$. Note that since rewards are binary and terminal the $Q$-value at any state $\in [0, 1]$.

- Assumption D.2 is a milder form of the typical assumption on the coverage over states visited by the optimal policy. Here, we only assume that there is an optimal policy that can fit the dataset $\mathcal{D}_{\text{off}}$ we collected. Typically the coverage assumption places a uniform bound on the likelihood ratio over the state distributions of the optimal policy and the current RL policy $(d^{\pi^\star}/d^\pi)$ as in (Chang et al., 2024).

- Assumption D.3 is necessary to ensure that the KL between the prefix generating policy (empirical distribution over $\mathcal{D}_{\text{off}}$) and the base policy $\pi^0$ is finite. If the size of the dataset $\mathcal{D}_{\text{off}} \to \infty$ and the samples in $\mathcal{D}_{\text{off}}$ are all drawn IID from a policy $\nu \in \Pi$, then this assumption requires that cross-entropy between $\nu$ and $\pi^0$ is finite.

**Assumption D.1** (Realizability of $Q$-function class). There is a finite $Q$-function class $\mathcal{F} \subseteq \{f : \mathcal{S} \times \mathcal{A} \to [0, 1]$, and that $Q$-function induced by any policy is realized in this class, i.e., $Q^\pi \in \mathcal{F} \, \forall \pi \in \Pi$.

**Assumption D.2** (Correctness and realizability of $\mathcal{D}_{\text{off}}$). We say that $\mathcal{D}_{\text{off}}$ is correct if it contains a single correct trace $\boldsymbol{y}$ for every $\boldsymbol{x} \in \mathcal{D}$ and realizable if $\exists$ some policy $\mu \in \Pi$ such that $\mu(\boldsymbol{y} \mid \boldsymbol{x}) = 1$, $\forall(\boldsymbol{x}, \boldsymbol{y}) \in \mathcal{D}_{\text{off}}$.

**Assumption D.3** (Bounded likelihood of $\mathcal{D}_{\text{off}}$ under $\pi^0$)**.** The KL divergence between base LLM $\pi^0$ and the policy $\mu \in \Pi$ that perfectly fits the data is $\text{KL}(\mu||\pi^0) < \infty$. In other words, this assumes that the samples in $\mathcal{D}_{\text{off}}$ have a bounded likelihood under the base LLM $\pi^0$, i.e., $\text{KL}(\mu||\pi^0) = \frac{1}{|\mathcal{D}_{\text{off}}|} \sum_{(\boldsymbol{x},\boldsymbol{y^x}) \in \mathcal{D}_{\text{off}}} -\log \pi^0(\boldsymbol{y^x} \mid \boldsymbol{x}) < \infty$.

*Proof.* We prove the guarantee against the comparator policy $\mu$ from Assumption D.2. Since $\mathcal{D}_{\text{off}}$ is correct and realizable by $\mu$ (i.e., for each $\boldsymbol{x} \in \mathcal{D}$ the unique correct trace $\boldsymbol{y^x}$ satisfies $\mu(\boldsymbol{y^x} \mid \boldsymbol{x}) = 1$), we have $J(\mu) = J^\star$. Thus it suffices to upper bound:

$$J(\mu) - J(\bar{\pi}_T). \tag{D.15}$$

**State distribution induced by $\mathcal{D}_{\text{off}}$.** Let $\boldsymbol{s} \in \mathcal{S}$ denote an autoregressive prefix-state. Algorithm 1 samples prefix-states by drawing $(\boldsymbol{x}, \boldsymbol{y})$ uniformly from $\mathcal{D}_{\text{off}}$ and then sampling a prefix of $\boldsymbol{y}$ (according to the algorithm's prefix-selection rule). Because $\mu$ deterministically generates the same trace $\boldsymbol{y}$ in $\mathcal{D}_{\text{off}}$ for each $\boldsymbol{x}$, this state distribution coincides with the state-visitation distribution of $\mu$; we denote it by $d_{\boldsymbol{s}}^\mu$.

$$d_{\boldsymbol{s}}^\mu \;\equiv\; d_{\boldsymbol{s}}^{\text{off}}. \tag{D.16}$$

Let $\pi^0$ be the base LLM and $\{\pi^t\}_{t=1}^T$ be the iterates produced by NPG / mirror descent with stepsize $\eta$ and critic $\widehat{Q}^t$, and define the averaged policy:

$$\bar{\pi}_T \;:=\; \frac{1}{T} \sum_{t=1}^T \pi^t. \tag{D.17}$$

**Performance difference lemma under $d_{\boldsymbol{s}}^\mu$.** Applying performance difference Lemma D.4 with $(\pi, \pi') = (\mu, \pi^t)$ yields:

$$J(\mu) - J(\pi^t) \;=\; \mathbb{E}_{\boldsymbol{s} \sim d_{\boldsymbol{s}}^\mu} \mathbb{E}_{\boldsymbol{a} \sim \mu(\cdot|\boldsymbol{s})} \big[A^{\pi^t}(\boldsymbol{s}, \boldsymbol{a})\big]. \tag{D.18}$$

Using $A^{\pi^t}(\boldsymbol{s}, \boldsymbol{a}) = Q^{\pi^t}(\boldsymbol{s}, \boldsymbol{a}) - V^{\pi^t}(\boldsymbol{s})$ and the identity $\mathbb{E}_{\boldsymbol{a} \sim \pi^t(\cdot|\boldsymbol{s})}[A^{\pi^t}(\boldsymbol{s}, \boldsymbol{a})] = 0$, (D.18) can be rewritten as

$$J(\mu) - J(\pi^t) \;=\; \mathbb{E}_{\boldsymbol{s} \sim d_{\boldsymbol{s}}^\mu} \Big[\big\langle Q^{\pi^t}(\boldsymbol{s}, \cdot), \, \mu(\cdot \mid \boldsymbol{s}) - \pi^t(\cdot \mid \boldsymbol{s}) \big\rangle \Big]. \tag{D.19}$$

### D.2.1. CRITIC ESTIMATION ERROR.

Fix an iteration $t$. The critic is fit by least squares over a finite class $\mathcal{F}$ (Assumption D.1) using $N$ i.i.d. samples $(\boldsymbol{s}_k, \boldsymbol{a}_k, z_k)$ where $\boldsymbol{s}_k \sim d_{\boldsymbol{s}}^\mu$, $\boldsymbol{a}_k \sim \rho^t(\cdot \mid \boldsymbol{s}_k)$ (see discussion below on $\rho^t$), and $z_k$ is an unbiased target for $Q^{\pi^t}(\boldsymbol{s}_k, \boldsymbol{a}_k)$. Because rewards are *terminal and binary* in $\{0, 1\}$, we have

$$0 \le Q^{\pi^t}(\boldsymbol{s}, \boldsymbol{a}) \le 1, \qquad 0 \le \widehat{Q}^t(\boldsymbol{s}, \boldsymbol{a}) \le 1, \tag{D.20}$$

so we may take $R = 1$ in Lemma D.6. Therefore, setting $\delta_t := \delta/(2T)$ and applying Lemma D.6 with $\mathcal{H} = \mathcal{F}$, with probability at least $1 - \delta_t$,

$$\mathbb{E}_{\boldsymbol{s} \sim d_{\boldsymbol{s}}^\mu, \, \boldsymbol{a} \sim \rho^t(\cdot|\boldsymbol{s})} \Big[\big(\widehat{Q}^t(\boldsymbol{s}, \boldsymbol{a}) - Q^{\pi^t}(\boldsymbol{s}, \boldsymbol{a})\big)^2\Big] \;\le\; \frac{256}{N} \log\Big(\frac{2|\mathcal{F}|}{\delta_t}\Big) \;=\; \frac{256}{N} \log\Big(\frac{4T|\mathcal{F}|}{\delta}\Big). \tag{D.21}$$

**Behavior distribution $\rho^t$ and pointwise domination.** At iteration $t$, Algorithm 1 forms critic data by first sampling $(\boldsymbol{s}_h, a_h^{\text{off}})$ uniformly from $\mathcal{D}_{\text{off}}$ and then sampling

$$a_h \;=\; \begin{cases} a_h^{\text{off}} & \text{w.p. } \frac{1}{2}, \\ a_h \sim \pi^t(\cdot \mid \boldsymbol{s}_h) & \text{w.p. } \frac{1}{2}. \end{cases} \tag{D.22}$$

Let $\mu(\cdot \mid \boldsymbol{s}_h)$ denote the (deterministic) conditional action distribution induced by $\mathcal{D}_{\text{off}}$, i.e.,

$$\mu(a \mid \boldsymbol{s}_h) \;:=\; \mathbf{1}\{a = a_h^{\text{off}}\}. \tag{D.23}$$

Then the induced action-sampling (behavior) distribution used for critic fitting is the mixture

$$\rho^t(\cdot \mid \boldsymbol{s}_h) := \frac{1}{2}\mu(\cdot \mid \boldsymbol{s}_h) + \frac{1}{2}\pi^t(\cdot \mid \boldsymbol{s}_h). \tag{D.24}$$

Consequently, for every state $\boldsymbol{s}$ and action $a$, we have the pointwise lower bounds

$$\rho^t(a \mid \boldsymbol{s}) \geq \frac{1}{2}\mu(a \mid \boldsymbol{s}), \qquad \rho^t(a \mid \boldsymbol{s}) \geq \frac{1}{2}\pi^t(a \mid \boldsymbol{s}), \tag{D.25}$$

and hence the pointwise domination inequalities

$$\mu(a \mid \boldsymbol{s}) \leq 2\rho^t(a \mid \boldsymbol{s}), \qquad \pi^t(a \mid \boldsymbol{s}) \leq 2\rho^t(a \mid \boldsymbol{s}). \tag{D.26}$$

In particular, (D.25) also implies absolute continuity: if $\rho^t(a \mid \boldsymbol{s}) = 0$ then $\mu(a \mid \boldsymbol{s}) = \pi^t(a \mid \boldsymbol{s}) = 0$, so $\mu(\cdot \mid \boldsymbol{s}) \ll \rho^t(\cdot \mid \boldsymbol{s})$ and $\pi^t(\cdot \mid \boldsymbol{s}) \ll \rho^t(\cdot \mid \boldsymbol{s})$.

Then Cauchy–Schwarz and Jensen applied to (D.21) yield, for $\pi \in \{\mu, \pi^t\}$,

$$\left| \mathbb{E}_{\boldsymbol{s} \sim d_{\boldsymbol{s}}^{\mu}, \, \boldsymbol{a} \sim \pi(\cdot \mid \boldsymbol{s})}\left[ \widehat{Q}^t(\boldsymbol{s}, \boldsymbol{a}) - Q^{\pi^t}(\boldsymbol{s}, \boldsymbol{a}) \right] \right| \leq \epsilon_{\text{crt}}, \qquad \epsilon_{\text{crt}} := 16\sqrt{2}\sqrt{\frac{1}{N}\log\left(\frac{4T|\mathcal{F}|}{\delta}\right)}. \tag{D.27}$$

Taking a union bound over $t \in [T]$, with probability at least $1 - \delta/2$, (D.27) holds for all $t$.

### D.2.2. MIRROR ASCENT AND NPG OPTIMIZATION ERROR.

The mirror-descent update at state $\boldsymbol{s}$ is the KL-regularized maximization

$$\pi^{t+1}(\cdot \mid \boldsymbol{s}) = \arg\max_{p(\cdot \mid \boldsymbol{s})}\left\{ \eta\langle \widehat{Q}^t(\boldsymbol{s}, \cdot), p(\cdot \mid \boldsymbol{s})\rangle - \text{KL}(p(\cdot \mid \boldsymbol{s}) \| \pi^t(\cdot \mid \boldsymbol{s})) \right\}.$$

The first-order optimality condition implies that for any $p(\cdot \mid \boldsymbol{s})$,

$$\left\langle -\eta\widehat{Q}^t(\boldsymbol{s}, \cdot) + \nabla_r \text{KL}(r(\cdot \mid \boldsymbol{s}) \| \pi^t(\cdot \mid \boldsymbol{s}))\big|_{r=\pi^{t+1}}, \, p - \pi^{t+1} \right\rangle \geq 0. \tag{D.28}$$

Set $p = \mu(\cdot \mid \boldsymbol{s})$ and rearrange (D.28) to obtain

$$\eta\langle \widehat{Q}^t(\boldsymbol{s}, \cdot), \mu(\cdot \mid \boldsymbol{s}) - \pi^{t+1}(\cdot \mid \boldsymbol{s})\rangle \leq \left\langle \nabla_r \text{KL}(r \| \pi^t)\big|_{r=\pi^{t+1}}, \, \mu - \pi^{t+1} \right\rangle. \tag{D.29}$$

Apply the KL three-point identity (Lemma (D.62)) with $p = \mu(\cdot \mid \boldsymbol{s})$, $r = \pi^{t+1}(\cdot \mid \boldsymbol{s})$, and $q = \pi^t(\cdot \mid \boldsymbol{s})$ to rewrite the right-hand side:

$$\left\langle \nabla_r \text{KL}(r \| \pi^t)\big|_{r=\pi^{t+1}}, \, \mu - \pi^{t+1} \right\rangle = \text{KL}(\mu \| \pi^t) - \text{KL}(\mu \| \pi^{t+1}) - \text{KL}(\pi^{t+1} \| \pi^t). \tag{D.30}$$

Combining (D.29) and (D.30) gives

$$\eta\langle \widehat{Q}^t(\boldsymbol{s}, \cdot), \mu - \pi^{t+1}\rangle \leq \text{KL}(\mu \| \pi^t) - \text{KL}(\mu \| \pi^{t+1}) - \text{KL}(\pi^{t+1} \| \pi^t). \tag{D.31}$$

Using (D.20) and Pinsker's inequality, we bound the shift term

$$\eta\langle \widehat{Q}^t(\boldsymbol{s}, \cdot), \pi^{t+1} - \pi^t\rangle \leq \eta\|\widehat{Q}^t(\boldsymbol{s}, \cdot)\|_{\infty}\|\pi^{t+1} - \pi^t\|_1 \leq \frac{\eta^2}{2} + \text{KL}(\pi^{t+1} \| \pi^t), \tag{D.32}$$

where the last inequality uses $\|\widehat{Q}^t\|_{\infty} \leq 1$ and $\|\pi^{t+1} - \pi^t\|_1^2 \leq 2\text{KL}(\pi^{t+1} \| \pi^t)$. Adding (D.31) and (D.32) cancels $\text{KL}(\pi^{t+1} \| \pi^t)$ and yields

$$\eta\langle \widehat{Q}^t(\boldsymbol{s}, \cdot), \mu - \pi^t\rangle \leq \text{KL}(\mu \| \pi^t) - \text{KL}(\mu \| \pi^{t+1}) + \frac{\eta^2}{2}. \tag{D.33}$$

Taking expectation over $s \sim d_s^\mu$ and summing over $t = 1, \ldots, T$ gives

$$\frac{1}{T} \sum_{t=1}^{T} \mathbb{E}_{s \sim d_s^\mu} \Big[ \big\langle \widehat{Q}^t(s, \cdot), \, \mu - \pi^t \big\rangle \Big] \ \leq \ \frac{D_1}{\eta T} + \frac{\eta}{2}, \tag{D.34}$$

where

$$D_1 \ := \ \mathbb{E}_{s \sim d_s^\mu} \Big[ \mathrm{KL}\big( \mu(\cdot \mid s) \, \| \, \pi^1(\cdot \mid s) \big) \Big]. \tag{D.35}$$

Using $\pi^1 = \pi^0$ and the definition of $\mathrm{KL}(\mu \| \pi^0)$ under the $\mathcal{D}_{\mathrm{off}}$-induced state distribution (Assumption D.3), we identify

$$D_1 \ = \ \mathrm{KL}(\mu \| \pi^0). \tag{D.36}$$

### D.2.3. COMBINING CRITIC ERROR AND OPTIMIZATION ERROR.

Starting from (D.19), add and subtract $\widehat{Q}^t$:

$$\begin{aligned} J(\mu) - J(\pi^t) &= \mathbb{E}_{s \sim d_s^\mu} \Big[ \big\langle \widehat{Q}^t(s, \cdot), \, \mu - \pi^t \big\rangle \Big] \\ &\quad + \mathbb{E}_{s \sim d_s^\mu} \Big[ \big\langle Q^{\pi^t}(s, \cdot) - \widehat{Q}^t(s, \cdot), \, \mu - \pi^t \big\rangle \Big]. \end{aligned} \tag{D.37}$$

On the high-probability event where (D.27) holds for both $\pi = \mu$ and $\pi = \pi^t$, the critic-error term is bounded by

$$\mathbb{E}_{s \sim d_s^\mu} \Big[ \big\langle Q^{\pi^t} - \widehat{Q}^t, \, \mu - \pi^t \big\rangle \Big] \ \leq \ \Big| \mathbb{E}_{s \sim d_s^\mu, \, a \sim \mu} \big[ \widehat{Q}^t - Q^{\pi^t} \big] \Big| + \Big| \mathbb{E}_{s \sim d_s^\mu, \, a \sim \pi^t} \big[ \widehat{Q}^t - Q^{\pi^t} \big] \Big| \ \leq \ 2 \, \epsilon_{\mathrm{crt}}, \tag{D.38}$$

which introduces no extra factor of $H$.

Averaging (D.37) over $t = 1, \ldots, T$ and using (D.17), we obtain

$$J(\mu) - J(\bar{\pi}_T) \ \leq \ \frac{1}{T} \sum_{t=1}^{T} \mathbb{E}_{s \sim d_s^\mu} \Big[ \big\langle \widehat{Q}^t(s, \cdot), \, \mu - \pi^t \big\rangle \Big] \ + \ 2 \, \epsilon_{\mathrm{crt}}. \tag{D.39}$$

Applying (D.34) to the first term in (D.39) yields

$$J(\mu) - J(\bar{\pi}_T) \ \leq \ \frac{D_1}{\eta T} + \frac{\eta}{2} + 2 \, \epsilon_{\mathrm{crt}}. \tag{D.40}$$

Choose $\eta := \sqrt{\frac{2 D_1}{T}}$ to balance the first two terms in (D.40), giving

$$\frac{D_1}{\eta T} + \frac{\eta}{2} \ = \ \sqrt{\frac{2 D_1}{T}}. \tag{D.41}$$

Combining (D.36), (D.27), and (D.40), and recalling $J(\mu) = J^\star$, we conclude that with probability at least $1 - \delta$,

$$J(\pi^\star) - J(\bar{\pi}_T) \ = \ J^\star - J(\bar{\pi}_T) \ \leq \ \mathcal{O} \left( \sqrt{\frac{\mathrm{KL}(\mu \| \pi^0)}{T}} \ + \ \sqrt{\frac{1}{N} \log\Big( \frac{T |\mathcal{F}|}{\delta} \Big)} \right). \tag{D.42}$$

$\square$

### D.3. Proof of Proposition 3.4

Now, we prove our separation result in Proposition 3.4 that lower bounds the performance gap between standard RL and PrefixRL. Here, standard RL runs Algorithm 1 but now without any access to the off-policy dataset $\mathcal{D}_{\mathrm{off}}$. In each iteration, the critic training dataset $\mathcal{D}_t$ is now populated with $(s, a, r)$ tuples where both $s$ and $a$ are sampled from the current policy $\pi^t$. So, unlike PrefixRL we never sample the state or prefix from $\mathcal{D}_{\mathrm{off}}$. In this simple worst-case instance we present below there is a single trajectory in $\mathcal{D}_{\mathrm{off}}$ that is also what that the optimal policy samples with probability 1 and attains performance of $J(\pi^\star) = 1$.

*Proof.* We present (i) the MDP instance together with a choice of *base policy* that generates the off-policy trace, and then (ii) an exponential lower bound for standard on-policy RL without $\mathcal{D}_{\text{off}}$, and (iii) a horizon-independent (non-exponential) upper bound for PrefixRL with $\mathcal{D}_{\text{off}}$.

**MDP instance (hidden rewarding binary string) and base policy.** Fix a horizon $H$ and an unknown binary string $\boldsymbol{b} = (b_1, \ldots, b_H) \in \{0,1\}^H$. Let the state space be

$$\mathcal{S} = \{s_0, s_1, \ldots, s_H\}, \tag{D.43}$$

where $s_{h-1}$ encodes the first $h-1$ actions taken so far ($s_0$ is the start state). The action space is $\mathcal{A} = \{0,1\}$. Transitions are deterministic: from $s_{h-1}$, taking action $a_h \in \{0,1\}$ moves to $s_h$. The episode ends at $s_H$ with terminal reward

$$r = \mathbf{1}\{(a_1, \ldots, a_H) = (b_1, \ldots, b_H)\}. \tag{D.44}$$

Thus, exactly one length-$H$ action sequence earns reward $1$.

Let $\pi^\star$ be the deterministic policy that selects $b_h$ at $s_{h-1}$ for each $h \in [H]$. Then $J(\pi^\star) = 1$. For the PrefixRL part, we also choose a *base policy* $\mu$ and an off-policy dataset $\mathcal{D}_{\text{off}}$: we set $\mu := \pi^\star$ and let $\mathcal{D}_{\text{off}}$ contain the unique successful trajectory of $\mu$, equivalently the state–action pairs

$$\mathcal{D}_{\text{off}} = \{(s_{h-1}, b_h) : h \in [H]\}. \tag{D.45}$$

### D.3.1. EXPONENTIAL LOWER BOUND FOR STANDARD ON-POLICY RL (ALGORITHM 1 WITHOUT $\mathcal{D}_{\text{off}}$).

We analyze an on-policy variant of Algorithm 1 in which *there is no off-policy dataset*: at each iteration $t$, the algorithm samples $N$ full episodes only from its current policy $\pi^t$, observes only terminal rewards $r^{(t,i)} \in \{0,1\}$, fits a critic, and updates the policy. Let $\widehat{\pi}_T$ denote the (possibly randomized) policy output after $T$ iterations (so the total number of full episodes is $TN$). We prove that for any such algorithm, there exists an instance $\boldsymbol{b}$ for which the expected suboptimality gap is at least $1 - (TN + 1)2^{-H}$.

**Yao's minimax setup.** By Yao's minimax principle (Yao, 1977), it suffices to fix an arbitrary adaptive algorithm and analyze its expected performance when the instance is random:

$$\boldsymbol{b} \sim \text{Unif}(\{0,1\}^H), \tag{D.46}$$

and we write $\mathbb{P}_{\boldsymbol{b}}, \mathbb{E}_{\boldsymbol{b}}$ for probability/expectation over this draw (and over the algorithm's internal randomness).

**Per-rollout success probability is $2^{-H}$.** Fix any rollout index $(t,i)$. Condition on the full interaction history up to this rollout and on the algorithm's internal randomness. Under this conditioning, the action string $\boldsymbol{a}^{(t,i)} \in \{0,1\}^H$ is some random element (with an arbitrary distribution induced by the algorithm), while $\boldsymbol{b}$ remains uniform and independent. Therefore,

$$\mathbb{P}_{\boldsymbol{b}}\Big[r^{(t,i)} = 1 \mid \text{history}\Big] = \mathbb{P}_{\boldsymbol{b}}\Big[\boldsymbol{a}^{(t,i)} = \boldsymbol{b} \mid \text{history}\Big] = \sum_{\boldsymbol{a} \in \{0,1\}^H} \mathbb{P}\Big[\boldsymbol{a}^{(t,i)} = \boldsymbol{a} \mid \text{history}\Big] \cdot \mathbb{P}_{\boldsymbol{b}}[\boldsymbol{b} = \boldsymbol{a}] = 2^{-H}. \tag{D.47}$$

Taking expectation over the history yields the unconditional version:

$$\mathbb{P}_{\boldsymbol{b}}[r^{(t,i)} = 1] = 2^{-H}. \tag{D.48}$$

**Probability of ever seeing a reward-1 rollout.** There are exactly $TN$ rollouts in total. By a union bound and (D.48) we get the following upper bound on the probability of seeing a reward $1$ rollout across all $T$ steps,

$$\mathbb{P}_{\boldsymbol{b}}\Big[\exists\, t \leq T,\, i \leq N : r^{(t,i)} = 1\Big] \leq \sum_{t=1}^{T} \sum_{i=1}^{N} \mathbb{P}_{\boldsymbol{b}}[r^{(t,i)} = 1] = TN \cdot 2^{-H}. \tag{D.49}$$

**Bound the expected value of the returned policy.** Let $\boldsymbol{a} \sim \widehat{\pi}_T$ denote the length-$H$ string generated by rolling out $\widehat{\pi}_T$ from $s_0$. On instance $\boldsymbol{b}$, $J(\widehat{\pi}_T) = \mathbb{P}[\boldsymbol{a} = \boldsymbol{b}]$. We can decompose this reward on two events: whether the training interaction ever produced a reward-1 rollout or not:

$$\mathbb{E}_{\boldsymbol{b}}[J(\widehat{\pi}_T)] \leq \mathbb{P}_{\boldsymbol{b}}\Big[\exists\, t, i : r^{(t,i)} = 1\Big] \cdot 1 + \mathbb{P}_{\boldsymbol{b}}\Big[\forall\, t, i : r^{(t,i)} = 0\Big] \cdot \sup_{\widehat{\pi}_T} \mathbb{E}_{\boldsymbol{b}}\Big[J(\widehat{\pi}_T) \mid \forall\, t, i : r^{(t,i)} = 0\Big]. \tag{D.50}$$

On the event $\{\forall t, i : r^{(t,i)} = 0\}$, the algorithm has never observed the rewarding string. Each zero-reward rollout rules out *at most one* candidate string, namely the realized action string $\boldsymbol{a}^{(t,i)}$. Hence, conditioning on $\{\forall t, i : r^{(t,i)} = 0\}$ only implies that $\boldsymbol{b}$ is not in a set of at most $TN$ excluded strings. Under the prior $\boldsymbol{b} \sim \mathrm{Unif}(\{0,1\}^H)$, the posterior is uniform over the remaining candidates, so

$$\max_{\boldsymbol{a} \in \{0,1\}^H} \mathbb{P}_{\boldsymbol{b}}\Big[\boldsymbol{b} = \boldsymbol{a} \mid \forall t, i : r^{(t,i)} = 0\Big] \leq \frac{1}{2^H - TN}, \tag{D.51}$$

and therefore

$$\sup_{\widehat{\pi}_T} \mathbb{E}_{\boldsymbol{b}}\Big[J(\widehat{\pi}_T) \mid \forall t, i : r^{(t,i)} = 0\Big] = \sup_{\widehat{\pi}_T} \mathbb{E}_{\boldsymbol{b}}\Big[\mathbb{P}(\boldsymbol{a} = \boldsymbol{b} \mid \widehat{\pi}_T, \forall t, i : r^{(t,i)} = 0)\Big] \leq \frac{1}{2^H - TN}. \tag{D.52}$$

Substituting (D.49) and (D.52) into (D.50) yields

$$\mathbb{E}_{\boldsymbol{b}}[J(\widehat{\pi}_T)] \leq \mathbb{P}_{\boldsymbol{b}}\Big[\exists t, i : r^{(t,i)} = 1\Big] \cdot 1 + \mathbb{P}_{\boldsymbol{b}}\Big[\forall t, i : r^{(t,i)} = 0\Big] \cdot \frac{1}{2^H - TN} \leq TN \cdot 2^{-H} + \frac{1}{2^H - TN}. \tag{D.53}$$

In particular, whenever $TN \leq 2^{H-1}$ we have $\frac{1}{2^H - TN} \leq 2^{-(H-1)}$, and thus

$$\mathbb{E}_{\boldsymbol{b}}[J(\widehat{\pi}_T)] \leq TN \cdot 2^{-H} + 2^{-(H-1)} \leq (TN + 2) \, 2^{-(H-1)}. \tag{D.54}$$

**Suboptimality lower bound and fix an instance.** Since $J(\pi^\star) = 1$, we obtain from (D.53) the gap bound

$$\mathbb{E}_{\boldsymbol{b}}[J(\pi^\star) - J(\widehat{\pi}_T)] \geq 1 - (TN + 2) \, 2^{-(H-1)}. \tag{D.55}$$

By Yao's minimax principle (Yao, 1977), there exists a fixed hidden string $\boldsymbol{b}$ for which the same bound holds for the algorithm on that instance. This proves the exponential lower bound for standard on-policy RL without access to $\mathcal{D}_{\mathrm{off}}$.

D.3.2. HORIZON-INDEPENDENT (NON-EXPONENTIAL) UPPER BOUND FOR PREFIXRL.

We now analyze Algorithm 1 on the same instance *with $\mathcal{D}_{\mathrm{off}}$*. As discussed, the separation mechanism is that Algorithm 1 explicitly samples states from $\mathcal{D}_{\mathrm{off}}$, which are precisely the prefix states along the $\pi^\star$ trajectory, thereby forcing visitation of optimal-trajectory states.

**Choosing $\pi^0$.** Let $\pi^0$ denote the initialization policy of Algorithm 1, *i.e.* $\pi^1 = \pi^0$. We choose $\pi^0$ to be the uniform policy on $\{0,1\}$ at every state *(crucially, $\pi^0$ is not instance dependent)*:

$$\pi_0(0 \mid s) = \pi_0(1 \mid s) = \tfrac{1}{2} \qquad \text{for all } s \in \{s_0, \ldots, s_{H-1}\}. \tag{D.56}$$

Since $\mu = \pi^\star$ is deterministic on the states in $\mathcal{D}_{\mathrm{off}}$, the KL term in Theorem 3.1 is not exponential in $H$. Under the state-averaged convention used in Theorem 3.1,

$$\mathrm{KL}(\mu \| \pi_0) := \mathbb{E}_{s \sim d_s^\mu}\Big[\mathrm{KL}\big(\mu(\cdot \mid s) \| \pi_0(\cdot \mid s)\big)\Big] = \log 2, \tag{D.57}$$

while under the summed convention it is $H \log 2$; in either case it is not exponential in $H$.

**Invoking PrefixRL guarantee in Theorem 3.3).** All assumptions required by Theorem 3.3 hold on this instance with $\mu = \pi^\star$ and the corresponding $\mathcal{D}_{\mathrm{off}} = \{\boldsymbol{b}\}$. Therefore, applying Theorem 3.3 to Algorithm 1 yields (with probability at least $1 - \delta$) the bound

$$J(\mu) - J(\bar{\pi}_T) \leq \sqrt{\frac{2\mathrm{KL}(\mu \| \pi_0)}{T}} + \mathcal{O}\left(\sqrt{\frac{1}{N} \log\Big(\frac{T|\mathcal{F}|}{\delta}\Big)}\right), \tag{D.58}$$

where $\bar{\pi}_T$ is the iterate-averaged policy output by Algorithm 1. Since $\mu = \pi^\star$ on this instance, $J(\mu) = J(\pi^\star) = 1$, and (D.58) implies a non-exponential suboptimality bound for PrefixRL. In particular, with the choice (D.56) we have $\mathrm{KL}(\mu \| \pi_0) = \log 2$ (or $H \log 2$ under the summed convention), so the bound has no $2^{-H}$ term.

**Separation mechanism.** The on-policy exponential lower bound arises because, without $\mathcal{D}_{\mathrm{off}}$, an algorithm only observes nonzero reward if it guesses the entire length-$H$ string correctly in a single episode. In contrast, Algorithm 1 with $\mathcal{D}_{\mathrm{off}}$ repeatedly samples *states along the $\pi^\star$ trajectory* via $\sim \mathcal{D}_{\mathrm{off}}$ and trains at those states using the mixture distribution of Algorithm 1. Overall, this yields a non-exponential sample complexity, establishing a worst-case separation.

D.3.3. WORST-CASE SEPARATION RESULT BETWEEN STANDARD RL AND PREFIXRL.

The analysis in the above subsection can be viewed either for a fixed hidden string $\boldsymbol{b}$, or under a randomized instance distribution. In particular, let $\boldsymbol{b} \sim \text{Unif}(\{0,1\}^H)$, and note that under this randomization the off-policy dataset $\mathcal{D}_{\text{off}} = \mathcal{D}_{\text{off}}(\boldsymbol{b})$ also changes with $\boldsymbol{b}$ since it contains the unique successful trajectory $(s_{h-1}, b_h)_{h=1}^H$.

**Lower bound for standard on-policy RL under random $b$.** For any on-policy algorithm that runs for $T$ iterations with $N$ full episodes per iteration (so $TN$ total episodes) and does not have access to $\mathcal{D}_{\text{off}}$, the expected value of the suboptimality gap satisfies

$$\mathbb{E}_{\boldsymbol{b}}[J(\pi_{\boldsymbol{b}}^\star) - J(\widehat{\pi}_T)] \geq 1 - (TN + 2)\, 2^{-(H-1)}, \tag{D.59}$$

where $\pi_{\boldsymbol{b}}^\star$ denotes the optimal policy for instance $\boldsymbol{b}$ and $TN \leq 2^{H-1}$.

**Upper bound for PrefixRL under random $b$.** Now consider Algorithm 1 (PrefixRL) run on the same randomized instance, where $\mathcal{D}_{\text{off}} = \mathcal{D}_{\text{off}}(\boldsymbol{b})$ is provided to the algorithm. Choose the initialization $\pi_0$ to be the uniform policy (so that $\text{KL}(\mu_{\boldsymbol{b}} \| \pi_0)$ is not exponential in $H$), with $\mu_{\boldsymbol{b}} := \pi_{\boldsymbol{b}}^\star$ as the base policy that generates $\mathcal{D}_{\text{off}}(\boldsymbol{b})$. Invoking the previously proved Theorem 3.1 yields (with probability at least $1 - \delta$ over the algorithm's sampling)

$$\mathbb{E}_{\boldsymbol{b}}[J(\pi_{\boldsymbol{b}}^\star) - J(\bar{\pi}_T)] \leq \sqrt{\frac{2\, \mathbb{E}_{\boldsymbol{b}}[\text{KL}(\mu_{\boldsymbol{b}} \| \pi_0)]}{T}} + \mathcal{O}\left(\sqrt{\frac{1}{N}\log\left(\frac{T|\mathcal{F}|}{\delta}\right)}\right), \tag{D.60}$$

and for the uniform initialization $\pi_0$ we have $\mathbb{E}_{\boldsymbol{b}}[\text{KL}(\mu_{\boldsymbol{b}} \| \pi_0)] = \log 2$ under the state-averaged convention (or $H \log 2$ under the summed convention), which is not exponential in $H$.

**Applying Yao's minimax principle.** Equations (D.59) and (D.60) establish an *average-case* separation under the uniform distribution over instances $\boldsymbol{b}$ (with $\mathcal{D}_{\text{off}}$ coupled to $\boldsymbol{b}$ in the PrefixRL case). By Yao's minimax principle, this implies that there exists a fixed instance $\boldsymbol{b}$ (and the above choice of initialization $\pi_0$, *e.g.* uniform) for which the same separation holds for any randomized standard on-policy algorithm without $\mathcal{D}_{\text{off}}$ versus Algorithm 1 with $\mathcal{D}_{\text{off}}$.

$\square$

### D.4. Auxiliary Lemmas

**Lemma D.4** (Performance difference lemma; (Kakade & Langford, 2002))**.** *For all policies $\pi, \pi'$ and initial state distribution $\rho$,*

$$\mathbb{E}_{\boldsymbol{s}_0 \sim \rho}\left[V^\pi(\boldsymbol{s}_0) - V^{\pi'}(\boldsymbol{s}_0)\right] = \mathbb{E}_{\boldsymbol{s}_h \sim d_{\boldsymbol{s}}^\pi} \mathbb{E}_{a_h \sim \pi(\cdot|\boldsymbol{s}_h)}\left[A^{\pi'}(\boldsymbol{s}_h, a_h)\right]. \tag{D.61}$$

*Proof.* See proof of Lemma 6.1 in Kakade & Langford (2002). $\square$

**Lemma D.5** (Three-point identity for KL)**.** *Let $p, q, r$ be distributions on a common measurable space such that $p \ll r \ll q$ and all quantities below are finite. Then*

$$\text{KL}(p\|q) = \text{KL}(p\|r) + \text{KL}(r\|q) + \langle p - r, \ \nabla_r \text{KL}(r\|q)\rangle, \tag{D.62}$$

*. For discrete distributions $p, q, r$ we have:*

$$\nabla_r \text{KL}(r\|q)(x) = \log\frac{r(x)}{q(x)} + 1. \tag{D.63}$$

*Equivalently we can state this the three-point identity for discrete distributions as:*

$$\text{KL}(p\|q) = \text{KL}(p\|r) + \text{KL}(r\|q) + \left\langle p - r, \ \log\frac{r}{q}\right\rangle, \tag{D.64}$$

*since $\langle p - r, 1\rangle = \int(p - r) = 0$ (or $\sum_i(p_i - r_i) = 0$).*

*Proof.* We work in the continuous case; the discrete case is identical with integrals replaced by sums. Recall

$$\mathrm{KL}(a\|b) = \int a(x)\log\frac{a(x)}{b(x)}\,dx.$$

Consider the difference between the left-hand side and the first two KL terms:

$$\begin{aligned}
\mathrm{KL}(p\|q) - \mathrm{KL}(p\|r) - \mathrm{KL}(r\|q) &= \int p\log\frac{p}{q}\,dx - \int p\log\frac{p}{r}\,dx - \int r\log\frac{r}{q}\,dx \\
&= \int p\left(\log\frac{p}{q} - \log\frac{p}{r}\right)\,dx - \int r\log\frac{r}{q}\,dx \\
&= \int p\log\frac{r}{q}\,dx - \int r\log\frac{r}{q}\,dx \\
&= \int (p-r)\log\frac{r}{q}\,dx \\
&= \left\langle p-r,\ \log\frac{r}{q}\right\rangle.
\end{aligned}$$

This proves (D.64). To obtain the gradient form (D.62), note that for $\mathrm{KL}(r\|q) = \int r\log(r/q)\,dx$ the pointwise functional derivative with respect to $r$ is

$$\nabla_r \mathrm{KL}(r\|q)(x) = \log\frac{r(x)}{q(x)} + 1,$$

so

$$\langle p-r,\ \nabla_r\mathrm{KL}(r\|q)\rangle = \left\langle p-r,\ \log\frac{r}{q}\right\rangle + \langle p-r, 1\rangle.$$

Finally, $\langle p-r, 1\rangle = \int (p-r)\,dx = 1-1 = 0$, yielding (D.62). $\qquad\square$

**Lemma D.6** (Lemma 15 in Song et al. (2022))**.** *Fix any $R > 0$, $\delta \in (0,1)$, and assume we have a class of real-valued functions $\mathcal{H} : \mathcal{X} \to [-R, R]$. Suppose we have $K$ i.i.d. samples $\{(x_k, y_k)\}_{k=1}^K$ where $x_k \sim \rho$ and $y_k$ is sampled via the conditional probability $p(\cdot \mid x_k)$:*

$$y_k \sim p(\cdot \mid x_k) := h^\star(x_k) + \epsilon_k,$$

*where $h^\star \in \mathcal{H}$ and $\{\epsilon_k\}_{k=1}^K$ are independent random variables such that $\mathbb{E}[y_k \mid x_k] = h^\star(x_k)$. Additionally, suppose that $\max_k |y_k| \le R$ and $\max_x |h^\star(x)| \le R$. Then the least squares solution*

$$\hat{h} \leftarrow \arg\min_{h\in\mathcal{H}} \sum_{k=1}^K \big(h(x_k) - y_k\big)^2$$

*satisfies, with probability at least $1 - \delta$,*

$$\mathbb{E}_{x\sim\rho}\left[\big(\hat{h}(x) - h^\star(x)\big)^2\right] \le \frac{256R^2\log(2|\mathcal{H}|/\delta)}{K}.$$

The proof is the same as in Song et al. (2022) and thus is omitted here.

## E. Additional Experiment Details and Results on Back-Generalization

This appendix provides some additional discussion on the back-generalization phenomenon with more results and full details for our experiments in Section 4.

## E.1. Back-generalization from prefixed to no-prefix problems goes beyond stitching.

Consider a straightforward *stitching* argument as an explanation for back-generalization (Zhang et al., 2025b; Qu et al., 2026). According to this, the model simply learns to complete from the off-policy intermediate states better without updating the next-token distributions on no-prefix problems, but still improves performance on no-prefix problems when the model happens to sample the same intermediate states on its own. Note that this argument still holds in the tabular policy setting. In contrast, we find that back-generalization indeed influences next-token distributions on untrained states (no-prefix problems), which can only occur through favorable function approximation in LLMs.

## E.2. Hard problems used in the in-context back-generalization experiment

Section 4 studies a meta-learning style setting where the policy is trained on a *prefixed problem* consisting of an in-context example (a full solved hard problem) followed by a target hard problem. We compare transfer when the in-context hard problem is structurally similar to the target problem versus when it is unrelated (Figure 7). We use the following three problems:

> **Hard Problem P1 (Pass@16 for base model is 0.119 and is different from P3)**
>
> A sphere tangent to the $xy$-plane has center having $z$-coordinate $> 0$. If it is projected from $P = (0, b, a)$ to the $xy$-plane, it gives the conic section $y = x^2$. If $a = p/q$ for integers $p, q$ what is $p + q$? Answer: 3.

> **Hard Problem P2 (Pass@16 for base model is 0.074 and is similar to P3)**
>
> League has 30 teams (East 16, West 14). Inside each division everyone has played others once. If we add interleague games, what is smallest $k$ for which every team gets exactly $k$ games? Answer: 29.

> **Hard Problem P3 (Pass@16 for base model is 0.063)**
>
> Amongst 300 people, no one has more than $k - 1$ friends. What is the smallest $k$ for which it might be impossible to create some new friendships so that everyone has exactly $k$ friends? Answer: 151.

**Relatedness criterion (P2 is similar to P3; P1 is different from P3).** P2 and P3 are both naturally expressed as graph feasibility problems with degree constraints, and their solutions rely on reasoning about global constraints induced by local degree requirements (regularity, parity, and obstruction arguments). In contrast, P1 is a geometry and conic projection problem whose solution structure does not share this graph-theoretic scaffold. Figure 7 uses this controlled notion of similarity to separate two effects: when the prefix and suffix share a compatible representation (P2→P3), training on prefixed problems yields substantially stronger transfer than when they do not (P1→P3).

## E.3. Back-generalization under model-family mismatch

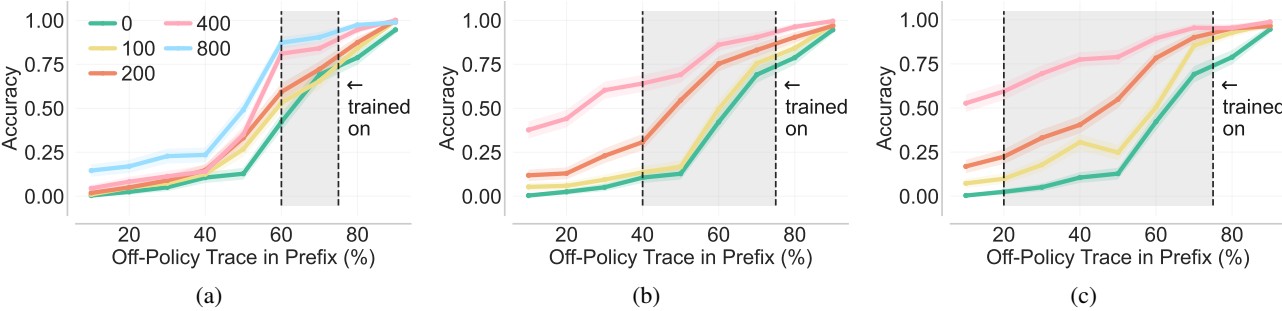

*Figure 14.* **Back-generalization (train-test mismatch):** On Llama3.1-8b-instruct we run RL only on prefixed problems sourced from Qwen3-4b-instruct whose prefix length (percent of the off-policy trace) lies in the shaded interval. At evaluation, we test across the full range of prefix lengths, including no-prefix problems (0% prefix). The performance at different RL training iterations (0, 100, 200, 400 and 800) is represented with different colors. Similar to Figure 4, where the prefixes are also sourced from Llama-3.1-8b When the mismatch is moderate, training on longer prefixes improves performance on shorter prefixes and can eventually improve no-prefix, indicating back-generalization (b, c). But different from Figure 4, we find that when the prefix distribution is skewed towards long prefixes back-generalization is weaker despite running RL for 800 iterations.

This section reports an additional train test mismatch experiment analogous to Figure 4, but where the off-policy prefixes

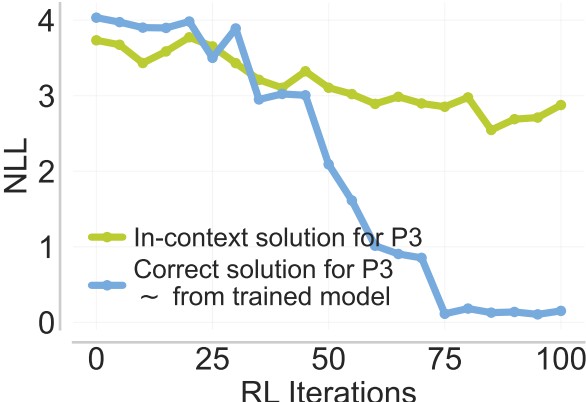

(a)

*Figure 15.* **(a) Likelihood of the in-context solution:** When asked to solve problem P3, we measure the negative log-likelihood (NLL) of the in-context solution for P3 provided in the context for P2, *i.e.*, when we run PrefixRL on P2 | P3 (setup in Section 4.3). Surprisingly, we find that the likelihood of the in-context trace drops less compared to a correct trace for P3 we sample from the final checkpoint. This indicates that back-generalization does not exactly clone the in-context prefix to improve performance on the no-prefix counterpart.

used to form prefixed problems are sourced from a different model family. Concretely, we train Llama3.1-8B-Instruct with PrefixRL while constructing prefixed problems from correct off-policy traces generated by Qwen3-4B-Instruct.

**Setup.** We first collect a pool of correct off-policy traces from Qwen3-4B-Instruct. Each trace induces a family of *prefix states* by truncating the trace to a specified prefix length (reported as a percentage of the full trace). A prefixed problem is formed by conditioning the training policy on such a prefix state and then asking it to complete the solution for the same underlying problem. During training, we restrict prefix lengths to lie in a band (the shaded interval in Figure 14), and we run on-policy RL *only* on these prefixed problems. At evaluation, we sweep prefix lengths across the full range, including the no-prefix endpoint (0% prefix), and report accuracy at multiple RL training iterations.

**Cross-family prefixes can still induce back-generalization.** Figure 14 shows that cross-family prefixes can still induce back-generalization, but the effect is less robust than in the same-family setting of Figure 4. When the mismatch is moderate (training includes prefixes that are not concentrated exclusively at the very end of traces), improvements appear near the trained prefix region and then propagate to shorter prefixes, eventually improving no-prefix performance (Figure 14b,c). In contrast, when the training prefix distribution is skewed toward long prefixes, back-generalization is weaker, and the transfer to shorter prefixes and to no-prefix remains limited even after long training (800 iterations) (Figure 14a).

**Choosing the right prefix distribution.** A key difficulty in this setting is that the training prefixes are not sampled from the current policy. They are injected from an external model (Qwen), and therefore correspond to states that can be very unlikely under the evolving Llama policy. When training concentrates on very late prefixes, the policy can improve primarily on a narrow slice of the prefixed state distribution without sufficiently shaping behavior on earlier states that dominate no-prefix rollouts. This makes cross-family prefixing most effective when the training mixture covers a sufficiently broad range of prefix lengths, rather than concentrating only on long, heavily conditioned prefixes.

### E.4. Mental Model: Function Approximation and Back-Generalization.

**NLL of the in-context solution changes little.** A natural hypothesis is that improvements on the untrained but in-context problem P3 come from memorizing the in-context trace and replaying it at test time. Instead, because this trace is extremely unlikely and never directly trained on, the model does not learn to imitate it (we also saw an example of this in Figure 5(b)). In Figure 15, the negative log-likelihood of the in-context P3 solution barely decreases under RL on P2|P3; the final policy instead prefers a different token sequence that still yields the correct answer. Together with the correlation in Figure 5, this suggests strong similarity between prefixed and no-prefix solutions, but weak similarity to the specific off-policy prefix used in the prefixed problem.

Although the model does not clone the in-context off-policy trace, performance still transfers to the in-context problem. Our speculation is that for long chain-of-thought rollouts, "state" is better viewed as the model's internal representation induced by the prefix: because the model self-corrects and backtracks, many distinct token sequences can map to similar latent states. Thus, while solving the prefixed problem, the policy can backtrack into representations close to those encountered when solving the original problem directly, but now with positive reward. If the history-conditioned and non-history-conditioned

representations are close, rewards observed in the former will shift the next-token distribution in the latter. This explains the overlap between prefixed and no-prefix responses (Figure 5), why transfer is stronger for related pairs like P2|P3 than unrelated ones like P2|P1 (Figure 7), and why the model still may not learn to reproduce the literal off-policy prefix when its rephrased representation is far from the original context (Figure 14(a)).

# F. Additional Experiments and Details for Results in Section 5

## F.1. Implementation details for PrefixRL and Baselines

### F.1.1. Off-policy RL Baselines

In our work, we use CISPO (Chen et al., 2025a) to compute the policy gradient following Khatri et al. (2025) which found it to work best on long RL runs. Moreover, CISPO can also handle off-policy updates, *i.e.*, update $\pi^t$ on a trajectory $\tau$ sampled from $\mu \neq \pi^t$ with an importance weighting term common in off-policy RL (Fujimoto et al., 2018). For each $\boldsymbol{x}$ in batch $\mathcal{B}$, CISPO samples $k$ reponses $\{\boldsymbol{y}_i^{\boldsymbol{x}}\}_{i=1}^k$ where $\boldsymbol{y}_i^{\boldsymbol{x}} \sim \mu(\cdot \mid \boldsymbol{x})$, then the CISPO policy gradient is given by:

$$\textbf{(CISPO)} \quad \frac{1}{\text{token-sum}} \sum_{\boldsymbol{x} \in \mathcal{B}} \sum_{i=1}^k \sum_{h=1}^{|\boldsymbol{y}_i^{\boldsymbol{x}}|} \left( \text{stop-grad} \left( \max \left( w(\boldsymbol{x}, y_{i,h}^{\boldsymbol{x}}), \varepsilon_{\text{high}} \right) \right) \cdot A(\boldsymbol{x}, \boldsymbol{y}_i^{\boldsymbol{x}}) \cdot \nabla_\pi \log \pi^t(y_{i,h}^{\boldsymbol{x}} \mid \boldsymbol{x}, \boldsymbol{y}_{i,<h}^{\boldsymbol{x}}) \right),$$

$$\text{where} \quad w(\boldsymbol{x}, y_{i,h}^{\boldsymbol{x}}) = \frac{\pi^t(y_{i,h}^{\boldsymbol{x}} \mid \boldsymbol{x}, \boldsymbol{y}_{i,<h}^{\boldsymbol{x}})}{\mu(y_{i,h}^{\boldsymbol{x}} \mid \boldsymbol{x}, \boldsymbol{y}_{i,<h}^{\boldsymbol{x}})} \quad \text{and} \quad \text{token-sum} = \sum_{\boldsymbol{x} \in \mathcal{B}} \sum_{i \in [k]} |\boldsymbol{y}_i^{\boldsymbol{x}}|. \tag{F.1}$$

In (F.1), the advantage $A(\boldsymbol{x}, \boldsymbol{y}_i^{\boldsymbol{x}})$ is computed by subtracting the baseline $\bar{r}(x) = \frac{1}{k} \sum_{i=1}^k r(\boldsymbol{x}, \boldsymbol{y}_i^{\boldsymbol{x}})$ from $r(\boldsymbol{x}, \boldsymbol{y}_i^{\boldsymbol{x}})$. The per-token importance weight $w(\boldsymbol{x}, y_{i,h}^{\boldsymbol{x}})$ accounts for the distribution shift between the current policy $\pi^t$ and the sampler $\mu$ and is 1 for on-policy traces where $\pi^t = \mu$. To reduce gradient variance from importance weights, it is clipped at $\varepsilon_{\text{high}}$ and in practice we set it to a value of $0.01$.

In our setup, the off-policy dataset $\mathcal{D}_{\text{off}}$ is constructed by *rejection sampling* the base policy $\pi^0$: for each prompt $\boldsymbol{x}$, we repeatedly sample $\boldsymbol{y} \sim \pi^0(\cdot \mid \boldsymbol{x})$ until we obtain one *correct* trajectory (according to the verifier), and store that successful trajectory in $\mathcal{D}_{\text{off}}$. This procedure induces an *accepted* (conditional) behavior distribution

$$\mu_{\text{acc}}(\boldsymbol{y} \mid \boldsymbol{x}) = \pi^0(\boldsymbol{y} \mid \boldsymbol{x}, r(\boldsymbol{x}, \boldsymbol{y}) = 1) = \frac{\pi^0(\boldsymbol{y} \mid \boldsymbol{x}) \, \mathbf{1}\{r(\boldsymbol{x}, \boldsymbol{y}) = 1\}}{a(\boldsymbol{x})}, \qquad a(\boldsymbol{x}) := \Pr_{\boldsymbol{y} \sim \pi^0(\cdot \mid \boldsymbol{x})} \left[ r(\boldsymbol{x}, \boldsymbol{y}) = 1 \right]. \tag{F.2}$$

Thus, when we treat accepted trajectories as "samples from $\mu$" in (F.1), the correct sequence-level importance ratio for an accepted trajectory $\boldsymbol{y}$ is

$$\frac{\pi^t(\boldsymbol{y} \mid \boldsymbol{x})}{\mu_{\text{acc}}(\boldsymbol{y} \mid \boldsymbol{x})} = a(\boldsymbol{x}) \cdot \frac{\pi^t(\boldsymbol{y} \mid \boldsymbol{x})}{\pi^0(\boldsymbol{y} \mid \boldsymbol{x})} \qquad (\text{since } \mathbf{1}\{r(\boldsymbol{x}, \boldsymbol{y}) = 1\} = 1 \text{ for accepted } \boldsymbol{y}). \tag{F.3}$$

The key point is that the acceptance-probability correction $a(\boldsymbol{x})$ is a *trajectory-level* factor: it appears once per accepted sequence, not once per token. In practice, we estimate $a(\boldsymbol{x})$ directly from the rejection-sampling effort. Let $R(\boldsymbol{x})$ be the number of rollout attempts required to obtain one correct trace for $\boldsymbol{x}$ during dataset construction; then $\widehat{a}(\boldsymbol{x}) \approx 1/R(\boldsymbol{x})$. CISPO, however, uses per-token importance weights $w(\boldsymbol{x}, y_{i,h}^{\boldsymbol{x}})$ (Eq. (F.1)) and aggregates gradients across tokens. A simple way to incorporate the acceptance correction is to multiply *each token* in an accepted trajectory by $\widehat{a}(\boldsymbol{x})$:

$$\widetilde{w}(\boldsymbol{x}, y_{i,h}^{\boldsymbol{x}}) := \widehat{a}(\boldsymbol{x}) \cdot \frac{\pi^t(y_{i,h}^{\boldsymbol{x}} \mid \boldsymbol{x}, \boldsymbol{y}_{i,<h}^{\boldsymbol{x}})}{\pi^0(y_{i,h}^{\boldsymbol{x}} \mid \boldsymbol{x}, \boldsymbol{y}_{i,<h}^{\boldsymbol{x}})}. \tag{F.4}$$

This heuristic is slightly biased relative to the sequence-level ratio in (F.3): multiplying every token effectively scales an accepted trajectory's total contribution by approximately $\widehat{a}(\boldsymbol{x}) \cdot |\boldsymbol{y}|$ (modulo the global token normalization), whereas the exact correction would apply $\widehat{a}(\boldsymbol{x})$ once at the trajectory level. When accepted trajectories have heterogeneous lengths, this introduces a mild length-dependent bias. Empirically, we found this approximation to be stable, and it preserves the intended qualitative effect: prompts that are harder under $\pi^0$ (larger $R(\boldsymbol{x})$, smaller $\widehat{a}(\boldsymbol{x})$) receive smaller off-policy gradient mass, reflecting the fact that an accepted sample from $\mu_{\text{acc}}$ is "more selective" for those prompts.

**LUFFY baseline: mixed-policy GRPO with policy shaping.** We also compare against LUFFY (Yan et al., 2025), which incorporates off-policy reasoning traces by mixing them with on-policy rollouts inside a GRPO-style objective. LUFFY

computes advantages using *group computation* over a mixed set of rollouts: for each prompt it combines $N_{\text{on}}$ on-policy samples with $N_{\text{off}}$ off-policy traces, and normalizes rewards using the mean and standard deviation over the union of the two groups. In our reproduction, we follow LUFFY's "fair" setting by using 8 total samples per prompt with a 1-off-policy / 7-on-policy split, rollout batch size 128, update batch size 64, and rollout temperature 1.0. For optimization, LUFFY uses a constant learning rate of $10^{-6}$ and trains for 500 steps. We also removes KL regularization (setting $\beta = 0$) and uses an entropy-loss coefficient of 0.01. Finally, LUFFY introduces *policy shaping* via a regularized importance-sampling transformation controlled by a parameter $\gamma$, and for this we use $\gamma = 0.1$, chosen after sweeoing over $\{0.01, 0.1, 0.2\}$.

### F.1.2. HYPERPARAMETER DETAILS FOR PREFIXRL

We use the REINFORCE (Ahmadian et al., 2024) on-policy algorithm for PrefixRL and standard RL. For this, we set the training batch size of 128 with, a constant learning rate of $1 \times 10^{-6}$. We turn off any KL regularization and also disable entropy regularization (entropy coefficient 0). We also use a gradient clipping of 1.0. We set the sampling temperature for training at 1.0 for Llama3.1-8b-instruct and 0.7 for Qwen-3-4b-instruct. At test-time we sample with a temperature of 0.7 for both models, including the inference to collect data for rejection sampling. For all our PrefixRL runs we use $n = 8$ rollouts per prompt in the batch. We use the same for standard RL, and off-policy RL, except specified otherwise. In all our runs in Section 5 and Section 4 we run RL training for 400 iterations, except for our RL runs in Section 4.3 and Section 4.2 where run the training for 100 iterations. We set the maximum context length $H$ of the output trace to be 16384 for both the off-policy traces and during the RL run. We do not change this maximum limit for the prefixed problems, depsite the prefix itself possibly being 12k-14k tokens long in some cases. To account for such cases, we set the maximum sequence length to be 36000 tokens, though in practice the completions for the prefixed-problems with longer prefixes are typically much shorter and this limit of 36000 is hit very rarely.

Before we run RL, we finetune the Llama3.1-8b-instruct model on OpenThoughtsV3 (Guha et al., 2025). For this, we first filter the dataset to only retain responses of token length $< 24192$. Then, we run SFT for 5 epochs on this data at peak learning rate of $8e - 5$. We use a batch size of 512 traces per batch and a cosine learning rate schedule that has a linear warm up (for 10% of the total iterations) followed by a cosine decay to a learning rate of 8e-6. We use a hold out validation set to measure the negative log-likelihood loss during training, and pick the earliest checkpoint with the least validation loss as the final distilled model.

## F.2. Evaluation Protocols and FLOPs accounting.

All evaluation results in Section 5 are on the original no-prefix problems. For the plots where we report pass@$k$, we estimate it by drawing 256 samples per problem and using the bootstrapped estimate in (Chen et al., 2021). Where possible, we include 95% confidence intervals across evaluated problems. Details on FLOPs accounting for our compute-matched plots are as follows. We compute FLOPs using the standard Transformer approximation: processing $D$ tokens with a model of $N$ trainable parameters costs $\approx 2ND$ FLOPs for a *forward-only* pass (sampling/inference) and $\approx 6ND$ FLOPs for a *training update* (forward + backward + gradient computation) (Snell et al., 2024).

**Definitions.** We use $N$ to denote the number of trainable parameters of the model whose compute is being measured and $D$ to denote the total number of tokens processed by that model in the relevant stage, summed over all sequences (prompt + generated tokens).

**Per-iteration compute.** At RL iteration $t$, let $D_{\text{samp}}^{(t)}$ be the total number of tokens generated/evaluated during rollout sampling, and let $D_{\text{upd}}^{(t)}$ be the total number of tokens used in gradient-based optimization. We estimate FLOPs as

$$\text{FLOPs}^{(t)} = 2N\, D_{\text{samp}}^{(t)} + 6N\, D_{\text{upd}}^{(t)}. \tag{F.5}$$

**Cumulative compute.** The x-axis in our compute-matched plots reports cumulative FLOPs after $T$ iterations:

$$\text{FLOPs}_{\leq T} = \sum_{t=1}^{T} \text{FLOPs}^{(t)}. \tag{F.6}$$

If a method includes up-front rejection sampling to construct $\mathcal{D}_{\text{off}}$, we add that forward-only cost (also using $2ND$) to (F.6). So, if we need to sample $R$ times before we accept a correct trace for a problem $x$, then the total upfront compute spent on the problem is $2RND$.

### F.3. Qwen with Llama Prefixes

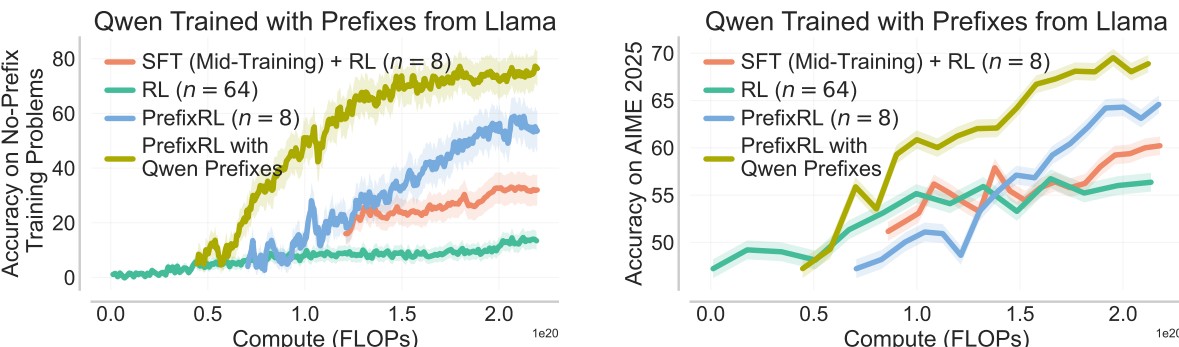

*Figure 16.* **Training Qwen model with prefixes from Llama.** Opposite of our experiment in Figure 9, here we train Qwen3-4b-instruct base LLM using off-policy traces sourced by rejection sampling Llama-3.1-8b-instruct model. We find that while off-policy prefixes from Llama are not as effective at improving Qwen as the other way around. Moreover, we also end up spending more compute upfront on rejection sampling the traces with the less capable Llama base model. We show accuracy on no-prefix training problems *(left)* and AIME 2025 *(right)*.

Figure 16 complements the cross-family back-generalization results in Figure 14 by reversing the direction of prefix transfer: instead of training Llama using Qwen-sourced off-policy prefixes, we train Qwen using prefixes sourced from Llama. The key takeaway is asymmetric transfer. While Qwen prefixes can still drive back-generalization when optimizing Llama (Section E.3), Llama prefixes are noticeably less effective for improving Qwen in both training accuracy (no-prefix training problems) and standardized evaluation (AIME 2025).

This asymmetry is consistent with the back-generalization discussion: PrefixRL relies on prefix states that are injected from an external distribution, and the degree to which learning transfers to no-prefix behavior depends on how informative and "compatible" those prefix states are with the target model's internal representations and solution strategies. When prefixes are sourced from a more capable model family (here, Qwen), they tend to encode higher-quality intermediate reasoning states, and RL can more readily propagate improvements from prefixed states to earlier states and ultimately to the no-prefix setting. In contrast, prefixes sourced from a less capable model (here, Llama) are both (i) harder to obtain via rejection sampling and (ii) less likely to contain strategy-revealing intermediate states that meaningfully constrain the continuation policy, resulting in weaker transfer when training Qwen.

A second practical implication is compute efficiency. Since PrefixRL amortizes training over a fixed pool of off-policy traces, the total compute depends not only on the on-policy RL phase but also on the *upfront* cost of harvesting correct traces. Rejection sampling from the weaker base model increases this upfront cost, and Figure 16 shows that even after paying that cost, the resulting prefixes yield smaller downstream gains for Qwen. Together with Figure 14, these results suggest that cross-family prefix sourcing is most attractive when (a) the source model is strong enough to produce diverse correct traces at reasonable cost, and (b) the injected prefix states align with the target model sufficiently well to allow back-generalization to propagate to the no-prefix distribution.

### F.4. PPO variant of PrefixRL.

Our main experiments instantiate PrefixRL with REINFORCE, since PrefixRL uses only on-policy rollouts during RL and REINFORCE avoids the additional memory overhead of maintaining a reference policy. To check that our empirical conclusions are not specific to this optimizer, we also evaluate a PPO variant of PrefixRL on top of the Qwen3-4B-Instruct base model. This experiment is also closer in spirit to our theoretical analysis with natural policy gradient, since PPO is commonly viewed as a scalable approximation to NPG for large-scale policy optimization.

We use the same training problems, off-policy prefix data, learning rate, batch size, and sampling configuration as in our REINFORCE experiments. As in our main compute-matched plots, we include the upfront cost of rejection sampling when plotting PrefixRL. Figure 17 shows that PPO exhibits the same qualitative behavior as REINFORCE: PrefixRL makes substantially faster progress on the original no-prefix hard problems than standard PPO-RL and mid-training on off-policy traces followed by PPO-RL. At matched compute, PrefixRL improves no-prefix hard-problem accuracy by more than 30% absolute over the strongest PPO baseline. This suggests that the gains from PrefixRL are not an artifact of the particular policy-gradient estimator used in our main experiments, but instead come from the prefixed-problem construction itself.

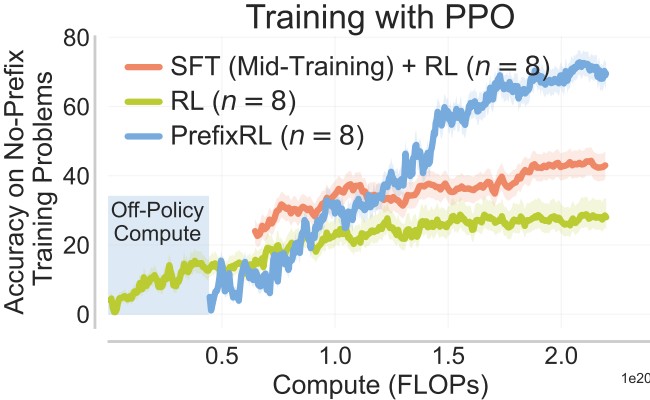

*Figure 17.* **PrefixRL improves compute-matched PPO training.** We evaluate a PPO variant of PrefixRL on Qwen3-4B-Instruct and compare against standard PPO-RL and SFT mid-training on off-policy traces followed by PPO-RL. PrefixRL includes the upfront rejection-sampling cost in the compute axis, shown as off-policy compute. Despite this offset, PrefixRL substantially outperforms both baselines on the original no-prefix hard training problems, indicating that PrefixRL's gains are not specific to REINFORCE and instead come from training on prefixed problems.

## F.5. Computing the gradient norm and standard deviation metrics

To quantify training signal-to-noise, we track (i) the *norm of the expected gradient* and (ii) the *standard deviation of the sampled gradient* throughout RL training, using the same procedure described at the end of Section 5.

Let $g_t \in \mathbb{R}^N$ denote the (flattened) stochastic policy gradient computed at iteration $t$ from the current minibatch (including any on-policy and/or off-policy contributions, depending on the method). We maintain exponential moving averages (EMA) of the first and second moments *coordinate-wise*:

$$m_t = \beta m_{t-1} + (1-\beta)\, g_t, \tag{F.7}$$

$$v_t = \beta v_{t-1} + (1-\beta)\, (g_t \odot g_t), \tag{F.8}$$

where $\odot$ denotes elementwise multiplication and $\beta \in (0,1)$ is a fixed smoothing constant.

**Gradient norm.** We report the norm of the mean gradient estimate as

$$\text{GRADNORM}_t \;=\; \|m_t\|_2. \tag{F.9}$$

**Gradient standard deviation.** We estimate the (coordinate-wise) variance as $s_t = v_t - m_t \odot m_t$ and report

$$\text{GRADSTD}_t \;=\; \sqrt{\sum_{i=1}^N \max\{(s_t)_i,\, 0\}} \;=\; \sqrt{\text{tr}\!\left(\widehat{\text{Cov}}(g_t)\right)}. \tag{F.10}$$

Equivalently, this standard-deviation metric corresponds to estimating the trace of the gradient covariance matrix via first/second moments.

### F.5.1. CONSTRUCTING PREFIXED PROBLEMS: AN EXAMPLE

> **Original Problem (No-Prefix)**
>
> ```
> <|im_start|>user
> Consider the cube whose vertices are the eight points (x, y, z) for which each of x, y,
> and z is either 0 or 1 .  How many ways are there to color its vertices black or
> white such that, for any vertex, if all of its neighbors are the same color then it
> is also that color?  Two vertices are neighbors if they are the two endpoints of
> some edge of the cube.  Please reason step by step, and put your final answer within
> boxed.
> <|im_end|>
> ```

---

Prefixed Problem

---

< |im_start| >user
Consider the cube whose vertices are the eight points $(x, y, z)$ for which each of $x, y$, and $z$ is either 0 or 1 . How many ways are there to color its vertices black or white such that, for any vertex, if all of its neighbors are the same color then it is also that color? Two vertices are neighbors if they are the two endpoints of some edge of the cube. Please reason step by step, and put your final answer within boxed. < |im_end| > < |im_start| >assistant
<think> This is a complex or challenging question, and it is difficult to provide a direct and correct answer. I need to think about it. Well, so we have a cube with vertices at all 0-1 triples, so 8 vertices total. Each vertex has degree 3, right? Because in a cube, each vertex is connected to three others (differing in one coordinate). The problem says: color vertices black or white such that for any vertex, if all its neighbors are the same color, then it's also that color. So let's parse that condition.
Let's rephrase the condition as a logical implication to make it clearer. For a vertex v, let N(v) be its neighbors (3 neighbors). Let c(v) be color, say 0=white, 1=black for simplicity (though labels might not matter, just two colors). The condition is: if c(u) = c(w) for all u, w in N(v), then c(v) = c(u). So in other words, it's not allowed for a vertex to be a different color from all its neighbors when all neighbors are the same color. Wait, actually, "if all neighbors are same color, then it's that color" { so the only way this fails is if there's a vertex where all neighbors are, say, black, but the vertex is white. Similarly, all neighbors white but vertex black is bad.
So forbidden configurations: a vertex with 3 black neighbors being white, or 3 white neighbors being black. All other configurations are allowed? Let's check: if a vertex has neighbors of different colors, then the condition is vacuously true? Wait, no: the condition is a conditional. The antecedent is "all neighbors same color", so if the antecedent is false (i.e., neighbors are not all same color), then the implication is true regardless of the vertex color. Yes! That's key. So the only forbidden things are vertices where all three neighbors are white and the vertex is black, or all three neighbors are black and the vertex is white. For any other coloring (where for every vertex, either it has mixed neighbors or if it has monochromatic neighbors, it's the same color as those neighbors), it's okay.
So maybe another way: for each vertex, when we look at its neighborhood (the three adjacent vertices), if the neighborhood is monochromatic (all one color), then the vertex must be that color. If the neighborhood is not monochromatic (has both colors), no restriction on the vertex color (well, it's already colored, but the condition doesn't impose anything here).
So our goal is to count all 256 colorings satisfying this condition for every vertex.
Maybe it's helpful to model this as a graph. The cube graph Q3, 8 vertices, each degree 3. We need colorings f: V(Q3) → 0,1 such that for all v in V, if f(N(v)) is constant, then f(v) = f(N(v)). So equivalently, for all v, if f(N(v)) not in f(neighbor) for some neighbor (i.e., N(v) not monochromatic), then no problem; if N(v) monochromatic, then f(v) must equal that color.
Let me try to list all possible colorings or find a pattern. Maybe start with colorings that are "monochromatic": all black or all white. Let's check if these work.
All black: take any vertex, all neighbors are black, so condition says it must be black { which it is. Good. Similarly all white: same thing, all neighbors white, so vertex must be white { works. So 2 colorings here.
Now what about colorings with some black, some white. Let's think about smaller cubes? Wait, maybe not. Let's consider linear extensions or something, but maybe better to think about the structure of the cube.
Another idea: in graph theory, such conditions can sometimes be related to "fixed points" under some operation, but maybe not. Alternatively, let's consider the possible colorings by their "stability" or by what constraints they impose.
Wait, let's take an example. Suppose we have a single black vertex, say the origin (0,0,0). Then its neighbors are (1,0,0), (0,1,0), (0,0,1), all white (by assumption). Now check w = (1,0,0), which is white. What are its neighbors? (0,0,0) [black], (1,1,0), (1,0,1). So w has one black neighbor, two unknowns so far.

Let's say we color (1,1,0) and (1,0,1) white. Then w has neighbors: black, white, white { so not monochromatic, so condition is satisfied for w. Now check the other endpoints: (1,1,0) is white, neighbors: (1,0,0) [white], (0,1,0) [white], (1,1,1) [unknown]. If we color (1,1,1) black, then (1,1,0) has neighbors: white, white, black { not monochromatic, okay. (1,0,1) similar: neighbors white, white, black (at (1,1,1)), so okay. Now (1,1,1) is black, neighbors: (1,1,0) [white], (1,0,1) [white], (0,1,1) [unknown]. If we color (0,1,1) white, then (1,1,1) has neighbors white, white, white { all white, but it's black! Uh-oh, contradiction!

