# OpenReview forum: "Reuse your FLOPs: Scaling RL on Hard Problems by Conditioning on Very Off-Policy Prefixes"
_ICML.cc/2026/Conference — ICML 2026 regular_

### Official Review · Reviewer_rbiP · 2026-03-09

**Soundness:** 3
**Presentation:** 4
**Significance:** 2
**Originality:** 1
**Overall Recommendation:** 4
**Confidence:** 4

**Summary:**

The paper presents an off policy RL post training method for LLM reasoning. While on policy RL has been highly effective for LLM post training, it is often data inefficient because trajectories that do not receive reward are typically discarded. This paper proposes a well motivated off policy alternative to address that limitation. The method is backed by solid theoretical analysis, and the empirical results are generally promising.

**Compliance With Llm Reviewing Policy:**

Affirmed.

**Final Justification:**

The authors resolved my concerns, and I am happy to increase my score.

**Key Questions For Authors:**

See weaknesses above

**Limitations:**

Yes

**Strengths And Weaknesses:**

Strengths:
- The paper is clearly written and easy to follow, and its main idea is immediately understandable.
- The empirical evaluation is comprehensive, with strong experimental coverage and detailed ablations.
- The theoretical results are presented rigorously and provide useful justification for the proposed PrefixRL framework.

Weaknesses:
- The main weakness is limited novelty. The proposed method seems highly similar to BREAD [1], which also leverages prefixes in an off-policy RL setting and demonstrates consistent improvements. In particular, the idea of avoiding backpropagation through difficult traces also appears related.
- Off-policy RL for LLM reasoning is not new and has been an active area of research for some time. While the authors compare against LUFFY, an important comparison with CHORD [2] is missing.
- The paper has a natural connection to curriculum learning, especially through its framing of easy versus hard reasoning problems, but this connection is not discussed, and no comparison to curriculum-based methods is provided [3,4].


[1] BREAD: Branched Rollouts from Expert Anchors Bridge SFT & RL for Reasoning, arXiv: 2506.17211, NeurIPS 2025

[2] On-Policy RL Meets Off-Policy Experts: Harmonizing Supervised Fine-Tuning and Reinforcement Learning via Dynamic Weighting arXiv: 2508.11408, ICLR 2026

[3] Parashar et al., Curriculum Reinforcement Learning from Easy to Hard Tasks Improves LLM Reasoning arXiv: 2506.06632, ICLR 2026

[4] Self-Evolving Curriculum for LLM Reasoning arXiv: 2505.14970

---

> ### Author Rebuttal · Authors · 2026-03-31
>
> Thank you for the thoughtful feedback. To address your concerns, we added new experiments comparing PrefixRL to BREAD and CHORD. In a compute-matched evaluation, PrefixRL outperforms the stronger of these baselines by more than 20% absolute accuracy on hard training problems, and it also yields stronger hold-out accuracy on AIME 2025, HMMT 2025, and IMO-AnswerBench. We also want to clarify that beyond the PrefixRL method, we contribute theoretical guarantees on consistency and sample efficiency, along with an analysis of a newly identified back-generalization phenomenon. Please let us know if this addresses your concerns. We are happy to discuss further if any remain.
>
> - **PrefixRL vs. CHORD and BREAD on training**: https://imgur.com/a/SYetv2R
> - **PrefixRL vs. CHORD and BREAD on test**: https://imgur.com/a/GsEy18O
>
> > ### **New Experiment: Comparing PrefixRL with BREAD.**
>
> We have **added a direct comparison to BREAD** (links above) and find that PrefixRL performs substantially better in a compute-matched evaluation, improving accuracy on hard training problems by over 20% absolute, and it also outperforms BREAD on the downstream benchmarks AIME 2025, HMMT 2025, and IMO-AnswerBench. We instantiate BREAD as closely as possible within our setup: we use the same source traces, the same training set, and RL optimization configuration as PrefixRL. Following BREAD, we treat the final solution after the thinking trace as the hint and use its anchor search procedure to adaptively determine how much of the hint to reveal. We note that BREAD improves performance over mid-training and standard RL, but falls short of PrefixRL.
>
> **Distinct from BREAD**, a key advantage of PrefixRL is that it can prefix on very long thinking traces generated by the base model itself, not just short human-written hints. This places us in a genuine self-improvement setting, where progress does not depend on collecting external supervision. Moreover, these long self-generated traces appear to provide richer intermediate states for training, which in turn supports stronger back-generalization. Finally, as we explain in Appendix E.1, the back-generalization phenomena is a stronger driver of performance on no-prefix problems, compared to the stitching argument made in BREAD. In fact, we find evidence contrary to stitching: back-generalization influences the next-token distribution on no-prefix problems, despite only training on the prefixed ones.
>
> > ### **New Experiment: Comparing PrefixRL with CHORD.**
>
> We have **added a direct comparison to CHORD** (links above) and find that while CHORD improves over mid-training it is much worse than PrefixRL by >20% accuracy on training problems and >8-10% pass@1 across multiple test benchmarks. For a fair comparison, we SFT on the same off-policy correct traces that PrefixRL uses. For best results, following CHORD we implemented an annealing schedule on the SFT loss, decayed over 25% of the RL run (Sec 3.2), and used the reward shaping term $\phi$ (Eq 5 in Sec 3.3).
>
> > ### **Key contributions that further distinguishes PrefixRL from BREAD, CHORD and other baselines.**
>
> *Self-improvement setting*. PrefixRL operates in a pure self-improvement loop: the base LLM generates the off-policy traces, which are then reused through prefixing during later RL, enabling compute-matched comparisons that account for the cost of off-policy data generation. This is distinct from BREAD, CHORD, and related baselines, which don't operate in a self-improvement loop.
>
> *Theoretical guarantees*. We show that, under our assumptions, PrefixRL’s training objective remains consistent with the original no-prefix objective, different from approaches like CHORD where the SFT loss is directly added to the RL objective. Also, we provide finite-sample statistical learning guarantees for PrefixRL under partial coverage and general function approximation (gold standard setting in RL theory).
>
> *Back-generalization*. We identify back-generalization as a key phenomenon behind PrefixRL and design controlled experiments to study it empirically. These show that training only on prefixed problems improves the corresponding no-prefix problems as well, suggesting that PrefixRL does more than simply stitch together partial solutions.
>
> > ### **Connection to curriculum learning.**
>
> PrefixRL should be complementary to curriculum learning. PrefixRL doesn’t provide a curriculum but instead diversifies the difficulty of hard problems into a spectrum based on the length of the off-policy prefix. Curriculum learning could be applied to select the prefixed problems with the appropriate range of difficulties to further improve compute efficiency of PrefixRL, but we leave this to future work. Although, as BREAD uses the anchor search approach which acts as an adaptive curriculum, our new experiments provide at least one comparison against curriculum based approaches. Thank you for the great suggestion and we will add a discussion on this in the paper.

---

> > ### Author Rebuttal · Reviewer_rbiP · 2026-04-01
> >
> > > On comparisons with BREAD
> >
> > The authors emphasize that PrefixRL operates on self-generated "long thinking traces" rather than short human-written hints. However, this suggests that the underlying algorithmic framework is fundamentally identical to BREAD (the addition of being applied to a sparser reward regime?), with the primary distinction being data provenance. If the performance gains rely primarily on self-generated prefixes being more "in-distribution" with the base model, could the authors clarify if there are any core algorithmic innovations here, or if PrefixRL is an in-distribution, data-centric adaptation of BREAD?
> >
> > > Connection to Curriculum Learning
> >
> > The assertion that PrefixRL is "orthogonal" to curriculum learning is strong; it aligns closely with hint-based Curriculum Reinforcement Learning (CRL). By scaffolding generation with self-generated prefixes, PrefixRL effectively implements a reverse start-state curriculum, conceptually similar to Reverse Curriculum Generation[1]. Both BREAD and recent theoretical frameworks [2] classify this type of structural guidance as a hint-based curriculum.
> >
> > Therefore, while PrefixRL is orthogonal to data-based curriculum (modulating task difficulty or sample ordering), it fundamentally operates as a process-based curriculum. Given that [2] establishes theoretical guarantees for hint-based curricula, how do the theoretical contributions of PrefixRL build upon or diverge from [2]?
> >
> > > Summary
> >
> > I really enjoyed reading this paper. It has strong merits, especially in how cleverly it solves the difficult problem of reward sparsity in RLVR.
> >
> > For me, the true novelty of your work is not the basic algorithm of using prefixes, since that has been done before. The real contribution is using in-distribution, self-generated traces from the model itself. This feels like a smarter, more natural way to guide the model compared to BREAD, which relies on outside expert traces.
> >
> > I appreciate that you provided the comparison between the two methods in your response. However, my core concern remains unresolved. It is not clear to me when BREAD (using expert traces) might actually be better than your setup, and vice versa.
> >
> > I would love to see a detailed discussion on this, ideally supported by a few experiments.
> >
> > If provided, I would be more than happy to increase my store
> >
> >
> > [1] Florensa, C., Held, D., Wulfmeier, M., Zhang, M., & Abbeel, P. (2017). Reverse curriculum generation for reinforcement learning. In Conference on Robot Learning (CoRL) (pp. 482-495). PMLR.
> >
> > [2] Bu, D., Huang, W., Han, A., Nitanda, A., Wong, H.-S., Zhang, Q., & Suzuki, T. (2025). Provable Benefit of Curriculum in Transformer Tree-Reasoning Post-Training. arXiv preprint arXiv:2511.07372.

---

> > > ### Author Response · Authors · 2026-04-02
> > >
> > > Thank you for the reply. We appreciate that you “really enjoyed reading this paper”, and find that “it has strong merits, especially in how cleverly it solves the difficult problem of reward sparsity in RLVR.” We respond to the remaining concerns below by **adding two new experiments** which we plan to add to the final version of the paper.
> > >
> > > > “Are there any core algorithmic innovations here, or is PrefixRL an in-distribution, data-centric adaptation of BREAD? ... It is not clear to me when BREAD (using expert traces) might actually be better than your setup, and vice versa.”
> > >
> > > Beyond using self-generated off-policy traces, PrefixRL is fundamentally an RL acceleration algorithm (the PrefixRL objective is consistent with the original RL objective, and thus the off-policy data is not just a way to control the model behavior) while BREAD does not have this guarantee (but may be more useful for distilling off-policy behavior). This fundamental difference is due to the following algorithmic change. PrefixRL injects the off-policy prefix as if it were generated by the model itself, which places the model onto a state that is both realizable by the model (since it’s self generated) and can lead to a correct answer. BREAD conditions on the off-policy information (parts of the reference solution) as a hint but does not make the promise of placing the model onto a state that can be realized with on-policy sampling. We show this difference in **two new experiments**:
> > >
> > > **Single-example RL (same setting as Figure 5b)** Recall that in this setting, we run RL on a single training problem (P3 in App E.2) where the correct off-policy trace mentions the Erdos Gallai theorem. Here, we found that PrefixRL quickly learns to use the Dirichlet theorem instead of Erdos Gallai. When we run BREAD, the model does not change strategies from Erdos Gallai since the prompt asks the model to use the hint containing Erdos Gallai (link: https://imgur.com/a/OEJnktV). In contrast, PrefixRL injects the off-policy prefix as if it were part of a self-generated thinking trace, and the model is free to explore and consider it as a failed "in-context" attempt instead of a golden example. Here, we see that PrefixRL is not bound to follow the same strategy as the injected prefix, while BREAD is, which limits the exploration. This also results in a performance difference on the original problem since “Dirichlet Theorem” is a more robust way of solving it.
> > >
> > > **Back-generalization (same setting as Fig 4)** Here we ran PrefixRL with only prefixed problems (where the prefixes are a certain % of the off-policy trace) and found that it still generalizes well to no-prefix problems: when training only on prefixed-problems with 40%-80% length prefixes, at iteration 400 PrefixRL obtains 50% no-prefix accuracy (link: https://imgur.com/a/mWnwQHT). When we run BREAD in the same setting (hints using 40% to 80% of the reference solution on the same problems), we find that there is much less back-generalization: at iteration 400, the no-prefix accuracy is less than 25% (https://imgur.com/a/mWnwQHT). This difference comes from how PrefixRL places the model onto an off-policy state (part of a realizable self-generated trace) directly instead of giving the off-policy reference solution as a hint like BREAD.
> > >
> > > **Weaker back-generalization in hinted algorithms like BREAD** likely also results in a greater need for a curriculum from longer to shorter hints, as a more direct mechanism of improving performance on the original hard problem after learning to solve the hinted one.
> > >
> > >
> > >
> > > > “PrefixRL effectively implements a reverse start-state curriculum, conceptually similar to Reverse Curriculum Generation[1]”
> > >
> > > We clarify that **PrefixRL trains on the prefix lengths in a random order without a curriculum**. During the run, PrefixRL may receive positive rewards on more of the easier problems (longer prefixes) first before harder ones (shorter prefixes), but this is true for any RL run.
> > >
> > > We expect that adding a curriculum based on reward or prefix length could improve PrefixRL, since it’s likely that PrefixRL still wastes some compute early in training on some prefixes that are too short (too hard). In summary, PrefixRL creates data that could be used in a curriculum, but trains on the data in a random order. **PrefixRL works well even without a curriculum due to its strong back-generalization, but we expect that adding a curriculum would still benefit**.
> > >
> > > > How do the theoretical contributions of PrefixRL build upon or diverge from [2]?
> > >
> > > While the biggest difference is that PrefixRL does not use curriculum learning, the theory also differs from [2] because we use prefixes from realizable self-generated traces, rather than oracle prefixes. This lets us prove objective consistency with the original RL objective, while relying only on a weaker closeness condition between the base policy and the trace-generating policy, via a KL term, instead of worst-case stagewise state coverage.

---

### Official Review · Reviewer_5PAD · 2026-03-11

**Soundness:** 4
**Presentation:** 4
**Significance:** 3
**Originality:** 4
**Overall Recommendation:** 5
**Confidence:** 5

**Summary:**

This paper introduces PrefixRL, a framework designed to scale RL post-training for LLMs on difficult problems where the base model’s pass@k performance is extremely low, even for large values of 𝑘. In such cases, standard policy-gradient RL methods often fail to obtain any learning signal because correct trajectories are rarely discovered, causing the reward signal to plateau.

To mitigate this, PrefixRL utilizes off-policy correct traces that are sampled from the base model through large-scale rejection sampling (or that could have been collected from previous RL runs during training). Instead of using these off‑policy correct traces as supervision targets, the authors create a set of prefixed problems by appending a few fixed prefixes from off‑policy correct traces to the original problems and jointly optimize the policy on both prefixed and non‑prefixed problems in an on‑policy manner. These prefixed problems can be viewed as frontier states from which reaching the correct solution for the original problem becomes likely, allowing the policy to obtain a good learning signal.

The authors demonstrate the efficacy of PrefixRL in terms of compute efficiency and final performance compared to other baselines (on‑policy RL, off‑policy RL, and SFT+RL) on multiple math datasets for two “thinking” language models (Llama‑3.1‑8B‑Instruct distilled on the OpenThoughtsV3 dataset and Qwen‑3‑4B‑Instruct).

**Compliance With Llm Reviewing Policy:**

Affirmed.

**Final Justification:**

The authors have addressed my concerns, and I will maintain my positive score.

**Key Questions For Authors:**

Please refer to Weaknesses above.

**Limitations:**

yes.

**Strengths And Weaknesses:**

## Strengths
1. The paper is well written and easy to follow.
2. The paper has a good set of experiments and ablation study.
3. **Conceptual clarity and Novelty** - Reframes off policy traces not as supervision targets but as start-state distribution giving a clean on-policy formulation that still “reuses FLOPs” via conditioning on prefixes of off-policy correct traces.
4. **Theoretical grounding** - The authors provide a solid theoretical bridge by analyzing PrefixRL under Natural Policy Gradient (NPG), showing a tighter suboptimality bound with less samples than standard RL.

## Weaknesses
1. **Realizability of off‑policy traces** - For the proof of Theorem 3.2, the realizability assumption with a deterministic optimal policy is quite strong, especially since even the performance of PrefixRL on the training non‑prefix problems does not reach 100%. Can the proof be extended to a policy $\mu$ under which the off‑policy traces are realizable with probability at least $1-\delta$ ?
2. **Off‑policy dataset requirements** -  In the off‑policy dataset, is it necessary to have one correct trace for each original hard problem? There may be cases where, even with large‑scale rejection sampling, the base model never generates a correct trace for some hard problems. Also, does the off‑policy dataset have to remain static during the whole training process, or can you replace the off‑policy correct traces with correct rollout traces from previous RL steps? Doing so would make the off‑policy dataset highly realizable under the current policy. If PrefixRL adopt this strategy, how would it affect the training dynamics?
3. **Cross‑family prefixes for non‑thinking models** - Can cross‑family prefixes still induce back‑generalization for a non‑thinking model? In the paper, all experiments are done on two thinking models—Qwen‑3‑4B‑Instruct and Llama‑3.1‑8B‑Instruct distilled on OpenThoughtsV3 (where the correct responses are collected from a Qwen‑family model, QwQ‑32B). Thus, the off‑policy traces generated from both models may have a similar distribution, which might be one reason cross‑family generalization works. What would happen if the non‑distilled Llama‑3.1‑8B‑Instruct is trained with PrefixRL on off‑policy correct traces generated by Qwen‑3‑4B‑Instruct (where the prefixes are highly unlikely under the Llama base model)? Would back‑generalization still occur in this case?
4. **PrefixRL expands the set of solvable problems** (lines 324–328): The authors argue, based on Figure 8(c), that PrefixRL improves pass@32 accuracy and therefore expands the set of solvable problems. If Figure 8(a) shows pass@1 accuracy, then from the plots alone it is not entirely clear that there is additional improvement in performance when moving from pass@1 in Figure 8(a) to pass@32 in Figure 8(c).
5. Minor comments -
   - Typo: Line 345 – instead of “Figure 10(b)”, it should be “Figure 8(b)”.
   - Additional experiments on coding and planning domains would further validate the generality of the proposed approach beyond math reasoning.

---

> ### Author Rebuttal · Authors · 2026-03-31
>
> Thank you for the positive assessment of our work. To address your remaining concerns, we provide detailed answers to your questions. To improve the paper, we will update it with this discussion and fix any typos. Please let us know if this addresses all your concerns.
>
> >### **Realizability of off-policy traces (Assm 3.1). Can the proof be extended to a policy under which the off‑policy traces are realizable with probability?**
>
> We clarify that Assm. 3.1 only requires existence: there exists a policy in our class (i.e., there exists a set of weights in the transformer LLM architecture) that generates the correct traces. This assumption does not require that we can reach this policy through learning and optimization, so the empirical fact that the training performance of PrefixRL is below 100% does not contradict this assumption. This assumption mainly requires that our function class alone is sufficiently large, and is supported by universal approximation theorems that state, in principle, any continuous function can be approximated to any desired accuracy by neural networks.
>
> As you suggested, our proof can indeed be extended to approximate versions of this assumption (i.e., approximate realizability). In particular, if we relax the assumption to a policy that assigns high probability to correct traces, e.g., $\mu(y \mid x) \geq 1- \epsilon_x$, then the same proof holds with an additional approximation error term proportional to the weighted average of $\epsilon_x$. This is in the same spirit as approximation-error extensions in policy gradient theory [1]. In the sample complexity Thm 3.3, the approximation error will enter the KL term and scale according to an average of $- \log \pi_0(y \mid x)$. We will add this extension of our theoretical result to the paper.
>
> More broadly, we view (approximate) realizability as a standard and mild assumption in RL theory, which is substantially weaker than strong assumptions on the function class, such as Bellman-completeness [2,3].
>
> [1] Agarwal, Alekh, et al. On the theory of policy gradient methods: Optimality, approximation, and distribution shift. JMLR (2021).
>
> [2] Zhan, Wenhao, et al. Offline reinforcement learning with realizability and single-policy concentrability. COLT, 2022.
>
> [3] Chen & Jiang. Information-theoretic considerations in batch reinforcement learning. ICML, 2019.
>
>
> >### **Can cross‑family prefixes still induce back‑generalization for a non‑thinking model?**
>
> For our cross-family experiments, we restrict to thinking models mainly to retain the special formatting in thinking rollouts. If we use traces from a non-thinking model for prefixing, the thinking model would not try to complete the prefix and would instead start a fresh thinking block, making it harder for back-generalization to occur and for RL to make any progress. You also made an astute observation about Llama being effectively distilled on QwQ-32B traces. While this is true, our SFT is only light-weight mainly to pick up on “skills” like self-verification and back-tracking, as well as the thinking format. Studying the relationship between the prefix generator and the policy being trained more systematically is a promising direction for future work. In particular, it would be useful to understand how mismatch in reasoning style or trace format affects RL progress and back-generalization, and to explore trace formats beyond the current thinking format that may transfer more robustly across model families.
>
> >### **Off-policy dataset requirements: Is it necessary to have one correct trace for each original hard problem? Does data have to be static?**
>
> Having one correct off-policy trace per problem is just a simple design choice, not a requirement of PrefixRL. What is required is at least one correct trace for any problem on which we run PrefixRL; for the remaining problems, one can simply run standard RL.  You are also right that the off-policy data does not have to be static. We kept it fixed mainly for simplicity, but in principle the source of off-policy traces can be updated over time, for example using traces from later RL checkpoints. In fact, since PrefixRL can already use traces from different sources, including other model families, using later checkpoints should be quite feasible and may even reduce the degree of off-policyness. Thank you for the suggestion; we think this is a promising direction for future work.
>
> >### **Does PrefixRL expand the set of solvable problems (Fig 8)?**
>
> Yes, because PrefixRL improves Pass@32 (as we continue to train the Llama3.1-8b model) on training problems (Fig 8c). The correct pass@1 comparison for Fig 8c (Llama model pass@32) is Figure 2a, which shows the pass@1 accuracy of the Llama-based model. Here, we can see a gap between pass@1 and pass@32. To clarify, Fig 8a is the pass@1 accuracy of the Qwen-based model, so Fig 8a cannot be compared with Fig 8c. We apologize for the confusion and will make this clear in the final revision.

---

> > ### Author Rebuttal · Reviewer_5PAD · 2026-04-02
> >
> > Thanks for your responses. All of my major concerns have been resolved and I maintain my score.

---

> > > ### Author Response · Authors · 2026-04-04
> > >
> > > Thank you for the acknowledgement. We are very glad to know that our rebuttal has adequately addressed your concerns and that the issues are now "fully resolved". If so, we would greatly appreciate it if you would consider updating your initial score accordingly.

---

### Official Review · Reviewer_1gt2 · 2026-03-12

**Soundness:** 3
**Presentation:** 2
**Significance:** 2
**Originality:** 3
**Overall Recommendation:** 4
**Confidence:** 4

**Summary:**

This paper proposes PrefixRL, which reuses off-policy traces from rejection sampling as conditioning prefixes for on-policy RL. Conditioning on correct off-policy prefixes places the policy in high-rewarding states without the instabilities of importance weighting or the entropy collapse of SFT mid-training. The authors also claimed that their method has a back-generalization property. PrefixRL is shown to have a number of desirable properties and perform better than SFT-then-RL and other baselines.

**Compliance With Llm Reviewing Policy:**

Affirmed.

**Final Justification:**

The authors have addressed my concerns and I maintain my score.

**Key Questions For Authors:**

- What is the cost of rejection sampling? The paper defines "hard problems" as pass@512~0. For such problems, even pass@1 is vanishingly small. Rejection sampling until one correct trace requires hundreds or thousands of samples per problem. With 1k training problems, this could mean millions of traces upfront.

**Limitations:**

There is no discussion on limitations. For suggestions please see weaknesses.

**Strengths And Weaknesses:**

Strengths
- Using off-policy traces as conditioning context rather than supervision avoids distribution shift instabilities while still leveraging previously computed solutions.
- The theoretical analysis provides clean guarantees on the objective consistency and improved sample complexity under NPG.
- The experiments are quite thorough and the results are good.

Weaknesses
- Theorem 3.3 analyzes NPG, but the practical algorithm uses REINFORCE. The paper acknowledges this but the gap is substantial.
- Back-generalization is not theoretically explained. This is the most intriguing empirical finding, yet the mechanism remains a black box. Without understanding why back-generalization works, it's unclear when it will fail. In addition, this phenomenon is also present in many other post training methods.
- The related work mentions AdaBack, QuestA, etc., which also condition on partial solutions. These are natural baselines but are not benchmarked.
- Figures are extremely difficult to read

---

> ### Author Rebuttal · Authors · 2026-03-31
>
> Thank you for the positive assessment of our work, including that the "theoretical analysis provides clean guarantees", and that the "experiments are quite thorough and the results are good". To address your remaining concerns we add **new experiments**: 1) to match our practical implementation of PrefixRL with theoretical analysis; and 2) we compare PrefixRL with a new baseline BREAD that captures both AdaBack and QuestA. We answer your other questions below.
>
> >### **Theorem 3.3 analyzes NPG, but the practical algorithm uses REINFORCE.**
>
> We note that our main Theorem 3.2, which shows that PrefixRL shares the same global optimum as RL, already holds for REINFORCE, but Theorem 3.3 (sample complexity gains of PrefixRL) uses natural policy gradient (NPG). To address this gap, we add **new experiments** (link: https://imgur.com/a/MoJrDPH) where we use PPO (a known approximation for NPG [1]) for RL on qwen3-4b-instruct and get similar results to REINFORCE. For more details on this **please see our reply to Reviewer JzDu (2nd question)**.
>
>
> >### **Benchmarking PrefixRL against baselines that condition on partial solutions.**
>
> We add **new experiments** (links: https://imgur.com/a/SYetv2R & https://imgur.com/a/GsEy18O) comparing PrefixRL to BREAD [2], a strong baseline that conditions RL on partial hints and adaptively chooses how much of the hint to reveal via anchor search, capturing the core ideas behind both QuestA and AdaBack. In a compute-matched comparison, PrefixRL improves accuracy on hard training problems by over 20% absolute, while also outperforming BREAD on AIME 2025, HMMT 2025, and IMO-AnswerBench. Though BREAD does improve by 10% over our other baselines like standard RL and mid-training + RL. For a fair comparison, we use the same off-policy traces, training set, and RL optimization config as PrefixRL; we treat the solution after the thinking trace as the hint in BREAD, and account for the FLOPs spent on anchor search.
>
> **Distinct from BREAD, Questa, AdaBack**, PrefixRL uses self-generated thinking traces from the base model for self-improvement, rather than human-written hints or solutions. Our new experiment also suggests that longer self-generated traces provide richer intermediate states and lead to stronger back-generalization than prefixing only on the final solution. Finally, the curriculum aspect of AdaBack is reflected in our new BREAD experiments, which also use adaptive anchor search.
>
> [2] Zhang et al. BREAD: Branched rollouts from expert anchors.
>
> >### **What is the cost of rejection sampling?**
>
> **Our compute-matched plots includes rejection sampling cost**: For clarity, all of our compute-matched plots (Figs 2, 8 (a,c), and 9) already account for the rejection sampling compute, that is, we offset the PrefixRL run with the FLOPs used to collect the off-policy data. E.g., in Fig 2, the sampling compute is shown by the blue box preceding the PrefixRL curve, indicating that we spent 0.7×1e20 FLOPs on sampling the off-policy data (similarly in Figs. 8 and 9).
>
> **Details on the cost of rejection sampling**: For our experiments with Llama3.1-8B (Figure 2), 0.7×1e20 FLOPs corresponds to roughly 650 rejections sampled traces per training problem. Some problems require only about 150 traces, while others require more than 2000 before getting a correct trace (max limit of 2500). Overall, across 1000 training problems, we sampled about 650k traces during the rejection-sampling stage for Llama3.1-8B. **For comparison with training compute please see our response to Reviewer JzDu (1st question).** We will add these details to Sec 5 of the paper.
>
> >### **Theoretical analysis of back-generalization.**
>
> We provide a thorough empirical analysis in Sec 4 as a first step, and leave a theoretical or mechanistic explanation of back-generalization to future work. To probe its failure modes, Sec 4.3 shows that back-generalization occurs more readily through conditional distributions that are more likely under the model. In an in-context learning experiment, where we condition on either a related or unrelated problem and run RL only on a second appended problem, PrefixRL does not back-generalize through unrelated context.
>
> >### **Discussion of limitations for PrefixRL.**
>
> In Sec. 7, we note that a main limitation is that the mechanism behind back-generalization is still not fully understood, likely because it depends on how LLMs represent prefixes and intermediate states. Our experiments are also limited to math reasoning, so it remains unclear whether similar gains hold in code or other less structured domains. Finally, we study a relatively narrow class of off-policy data, mainly traces from the base model, though we also show strong results with traces from other models. Promising future directions include broader off-policy sources, such as agentic scaffold outputs, later RL checkpoints, and curriculum-based prefix selection. We will make these limitations more explicit in the final version.

---

> > ### Author Rebuttal · Reviewer_1gt2 · 2026-04-01
> >
> > The authors have addressed my concerns and I maintain my score.

---

> > > ### Author Response · Authors · 2026-04-04
> > >
> > > Thank you for the acknowledgement. We are very glad to know that our rebuttal has adequately addressed your concerns and that the issues are now "fully resolved". If so, we would greatly appreciate it if you would consider updating your initial score accordingly.

---

### Official Review · Reviewer_JzDu · 2026-03-13

**Soundness:** 3
**Presentation:** 4
**Significance:** 3
**Originality:** 4
**Overall Recommendation:** 4
**Confidence:** 4

**Summary:**

The paper proposes a novel approach for RL+LLM post-training, called PrefixRL, which augments the training dataset with additional "prefixed" prompts, which consist of training prompts and appended prefixes from correct off-policy responses. These prefixes allow on-policy RL training to benefit from off-policy data, particular rare correct responses to hard problems that otherwise might be rarely generated. The experiments instantiate this idea by rejection sampling from the base model before training, and randomly sampling prefixes from the correct base model responses. The results show that PrefixRL improves performance (measured by $pass@k$) over baselines like LUFFY, standard RL training, and SFT on correct responses followed by RL training, while also using less compute. The paper also describes 'back-generalization', an observed behavior where training only on data with added prefixes still improving performance on the original unprefixed problems.

**Compliance With Llm Reviewing Policy:**

Affirmed.

**Key Questions For Authors:**

For the rejection sampling stage, what was the average number of samples required to get a correct response? Was there an upper limit to how many samples you would take before stopping, or did you find correct responses for every prompt in this stage?

**Limitations:**

yes

**Strengths And Weaknesses:**

Strengths:
- The experimental validation on math is quite extensive. Both the performance of PrefixRL against reasonable baselines, and the relevance/existence of back-generalization are well-supported by the experimental results.
- The various ablation experiments, such as Figure 15 that shows the impact of the source of off-policy data on PrefixRL's performance, provide a clear picture of the effectiveness and limitations of PrefixRL and its sensitivity to different design choices.
- The paper proposes a method for improving compute efficiency in RL LLM post-training, which is a relevant and frequently encountered problem, particularly for large-scale training. The new approach is simple and well-supported by the experiments, and so is a significant contribution to an important problem.
- The paper is clearly written and easy to follow. The primary issue of wasted compute in post-training is well motivated, and the explanation of PrefixRL follows naturally from that.
- The most relevant works, both from classical RL, RL+LLM post-training, and RLHF are cited, and PrefixRL is clearly situated relative to these prior works
-  Conditioning on off-policy responses is a novel

Weakness:
-  Additional details on the rejection sampling procedure would be helpful. Implicit in the compute-matched results is that PrefixRL required fewer samples than online training, but extra information about the offline compute required (e.g. statistics about the number of responses required, ratio of rejection sampling compute to training compute, etc.) would be informative.
-   It would be good to state that the practical implementation of PrefixRL uses REINFORCE somewhere in the main text prior to the experiments. Currently it is stated in the appendix, and the only concrete instantiation of PrefixRL referenced in the main text is PrefixRL-NPG. This makes it easy to read through and associate the later experimental results with the theoretical NPG version of the algorithm, rather than the practical one.

---

> ### Author Rebuttal · Authors · 2026-03-31
>
> Thank you for the positive assessment of our work as a "significant contribution to an important problem", and for noting that the "experimental validation on math is quite extensive" and that "conditioning on off-policy responses is novel" approach. To address your remaining concerns: 1) we provide more details about the compute spent on rejection sampling of the off-policy data, in comparison to the training compute; and 2) we include **new experiments** to more closely match our practical implementation with theoretical analysis. Please let us know if this addresses all your concerns.
>
> >### **Additional details on the rejection sampling procedure would be helpful.**
>
> For clarity, all of **our compute-matched plots (Figures 2, 8 (a,c), and 9) already account for the compute spent on rejection sampling**, that is, we offset the PrefixRL run with the FLOPs used to collect the off-policy data. For example, in Figure 2, the rejection-sampling compute is shown by the blue box preceding the PrefixRL curve, indicating that we spent about 0.7×1e20 FLOPs on sampling the off-policy data (the same accounting is used in Figs. 8 and 9), before running PrefixRL on prefixes of correct traces in the off-policy data.
>
> **More details on the off-policy compute FLOPs**. For our experiments with Llama3.1-8B (Figure 2), the off-policy compute of 0.7×1e20 FLOPs corresponds to roughly 650 rejection sampled traces per training problem on average. Some problems require only about 150 traces to obtain a correct trace, while others require more than 2000 (we set a max limit of 2500 which a handful of problems exceeded). Note that we keep sampling until we get one correct trace and then immediately stop. Overall, across 1000 training problems, we sampled about 650k traces during the rejection-sampling stage for Llama3.1-8B. By comparison, PrefixRL training with the same base model runs for more than 400 iterations, with 8 traces per problem and a batch size of 128. Moreover, training on a trace is roughly 4x more expensive than sampling, once forward and backward passes are included. So, assuming similar trace lengths, the training stage roughly uses compute equivalent to at least 400×8×128×4 or about 1.64 million traces. Appendix F.2 has more details on the FLOPs accounting mechanism. Thank you for this great suggestion, we will add the above details to Section 5 of the paper to improve the discussion of our results in Fig 2.
>
>
> >### **It is easy to read through and associate the later experimental results with the theoretical NPG version of the algorithm, rather than the practical one that uses REINFORCE.**
>
> We mention that experiments use REINFORCE in L117-118 in Sec 3, but we will make it more clear in the final version. On the gap between theory and practice, we note that our main theoretical result (Theorem 3.2), which shows that PrefixRL shares the same global optimum as RL, already holds for REINFORCE, but Theorem 3.3 (sample complexity gains of PrefixRL) uses natural policy gradient (NPG); to address this gap, we add **new experiments** (link: https://imgur.com/a/MoJrDPH) where we use PPO (a known approximation for NPG [1]) for RL training done on top of the qwen3-4b-instruct base model, and get similar results to REINFORCE. This makes our practical implementation of PrefixRL closer to the algorithm we analyze in theory. Similar to our experiments with REINFORCE, we find that in a compute-matched plot, the performance of PrefixRL on the original hard problems (no-prefixes) is at least 30% better than the best baseline:  mid-training on off-policy data + RL with PPO.
>
> For the experiment with PPO, we used the same setting (e.g., learning rate, batch size) as our experiments for REINFORCE (details in App. F.1.2). Initially, we chose REINFORCE for its simplicity, since we are making purely on-policy updates, and thus can avoid loading a reference model in memory, to compute importance weights that PPO needs. We will add this new plot to the paper and also add more details in Sec 5 to make it clear earlier that we use REINFORCE for our main experiments (in addition to what we mention in L117-118 in Sec 3).
>
>
> [1] Chang et al. 2024 Dataset Reset Policy Optimization for RLHF
> Quote from the paper: “PPO can be understood as a heuristic that approximates NPG for the purpose of being more scalable for large-scale optimization”

---

> > ### Author Rebuttal · Reviewer_JzDu · 2026-04-06
> >
> > The authors have addressed my concerns and I maintain my score.

---

> > > ### Author Response · Authors · 2026-04-06
> > >
> > > Thank you for the acknowledgement. We are very glad to know that our rebuttal has "adequately addressed" your concerns and that the issues are now "fully resolved". If so, we would greatly appreciate it if you would consider updating your initial score accordingly. Thank you!

---

### Decision · Program_Chairs · 2026-04-30

**Decision:**

Accept (regular)

**Comment:**

This paper proposes PrefixRL, which reuses successful off-policy traces by conditioning RL on their prefixes rather than using them as supervision targets. The idea is simple, well motivated, and practically important for hard reasoning problems where standard on-policy RL has little learning signal. The paper combines a clean formulation, meaningful theoretical support, and strong empirical results on math reasoning.

The rebuttal addressed the main concerns well. The authors clarified that rejection-sampling cost is already included in the compute-matched plots, added more concrete cost statistics, and strengthened the theory-practice link with additional PPO experiments. They also added comparisons to BREAD and CHORD, which improves the novelty and evaluation case. Therefore, I recommend accept.

Reminder:
Reference: Nair, A., Srinivasan, P., Blackwell, S., Alcicek, C., Fearon, R., De Maria, A., Panneershelvam, V., Suleyman, M., Beattie, C., Petersen, S., Legg, S., and Mnih, V. Overcoming exploration in reinforcement learning with demonstrations. In 2018 IEEE International Conference on Robotics and Automation (ICRA), 2018. doi: 10.1109/ICRA.2018.8463167. arXiv:1709.10089.
Issue: authors mismatch with arXiv and IEEE